# KARPA: A Training-free Method of Adapting Knowledge Graph as References for Large Language Model's Reasoning Path Aggregation

## Abstract

Large language models (LLMs) demonstrate exceptional performance across a variety of tasks, yet they are often affected by hallucinations and the timeliness of knowledge. Leveraging knowledge graphs (KGs) as external knowledge sources has emerged as a viable solution, but existing methods for LLM-based knowledge graph question answering (KGQA) are often limited by step-by-step decision-making on KGs, restricting the global planning and reasoning capabilities of LLMs, or they require fine-tuning or pre-training on specific KGs. To address these challenges, we propose **Knowledge graph Assisted Reasoning Path Aggregation (KARPA)**, a novel framework that harnesses the global planning abilities of LLMs for efficient and accurate KG reasoning on KGs. KARPA operates through a three-step process: pre-planning, retrieving, and reasoning. First, KARPA uses the LLM's global planning ability to pre-plan logically coherent relation paths based on the provided question and relevant relations within the KG. Next, in the retrieving phase, relation paths with high semantic similarity to the pre-planned paths are extracted as candidate paths using a semantic embedding model. Finally, these candidate paths are provided to the LLM for comprehensive reasoning. Unlike existing LLM-based KGQA methods, KARPA fully leverages the global planning and reasoning capabilities of LLMs without requiring stepwise traversal or additional training, and it is compatible with various LLM architectures. Extensive experimental results show that KARPA achieves state-of-the-art performance in KGQA tasks, delivering both high efficiency and accuracy. Our code is available on `https://anonymous.4open.science/r/KARPA/`.

## 1 Introduction

In recent years, large language models (LLMs) (Touvron et al., 2023a;b; Achiam et al., 2023; Bai et al., 2023) have revolutionized natural language processing, demonstrating remarkable capabilities in understanding and generating human-like text across a range of tasks. Their ability to leverage vast amounts of data leads to impressive performance in areas such as information extraction (Xu et al., 2023), summarization (Jin et al., 2024), and question answering (Louis et al., 2024). However, these models face notable challenges, particularly in maintaining up-to-date knowledge, domain-specific knowledge (Zhang et al., 2024), or dealing with hallucinations (Zhang et al., 2023b; Huang et al., 2023) where the models produce incorrect or nonsensical outputs.

Knowledge graphs (KGs) present a promising solution to enhance the reasoning capabilities of LLMs by providing structured, reliable external knowledge (Zhu et al., 2024; Pan et al., 2024). Existing approaches that integrate LLMs with KGs generally fall into two categories. The first category involves direct interaction between the LLM and the KGs (Sun et al., 2023; Jiang et al., 2023), where the LLM explores the KG step-by-step. The second category, including methods such as reasoning on graphs (RoG) (Luo et al., 2023), involves generating retrieval information to extract knowledge from KGs. This often requires fine-tuning or pre-training the LLM on specific KG data (Li et al., 2023b; Huang et al., 2024). However, both approaches have notable limitations: (1) The direct interaction method often relies on local search strategies such as beam search, which can result

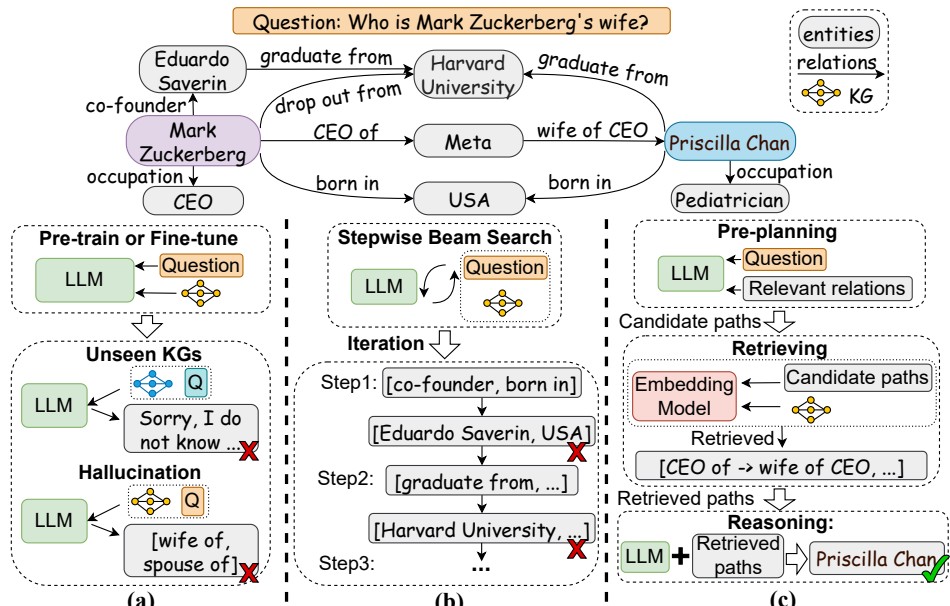

Figure 1: Comparison of different LLM-based KGQA methods: (a) Pre-training or fine-tuning the LLM for KGQA, which is prone to hallucinations and struggles to adapt to unseen KGs without extensive training process. (b) Direct reasoning over KGs using the LLM, which requires a high number of interactions between the LLM and KGs and is susceptible to local optima due to stepwise searching strategies. (c) Our KARPA framework, which leverages the global planning and reasoning capabilities of the LLM, enabling it to plan logically coherent relation paths based on all relevant relations within the KG. Our novel retrieval strategy allows the LLM to reason over complete relation paths, thus avoiding local optimal solutions while reducing interactions between the LLM and KGs.

in suboptimal answers by overlooking the LLM's potential for global reasoning and planning across the entire path. Moreover, this method typically demands a high number of interactions between the LLM and the KG, as illustrated in Figure 1(b). (2) In contrast, methods that involve pre-training or fine-tuning the LLM struggle with unseen KGs, often necessitating retraining. Additionally, they remain prone to hallucinations during the information generation process, as shown in Figure 1(a).

To address these limitations, we propose **Knowledge graph Assisted Reasoning Path Aggregation (KARPA)**, an innovative framework that leverages the global planning capabilities of LLMs alongside semantic embedding models for efficient and accurate KG reasoning. Our approach consists of three key steps: pre-planning, retrieving, and reasoning, as shown in Figure 1(c). In the pre-planning phase, KARPA enables the LLM to generate initial relation paths for the provided question using LLM's inherent reasoning and planning capabilities. With these inital relation paths, KARPA employs a semantic embedding model (Ruder et al., 2019) to identify candidate relations that are semantically similar to the relations within the initial paths. The LLM can then create coherent relation paths that logically connect the topic entity to potential answer entities using these candidate relations. During the retrieving phase, KARPA employs an embedding model to identify candidate paths within the KG that exhibit the highest similarity to the relation paths generated by the LLM in the pre-planning phase. This avoids locally optimal issues encountered in previous methods. Finally, during the reasoning step, the candidate paths and their corresponding tail entities are provided to the LLM to formulate final answers. The detail of our framework is shown in Figure 2.

KARPA offers several key advantages over existing LLM-based KGQA methods: (1) KARPA fully exploits the global planning and reasoning abilities of LLMs, generating comprehensive relation paths without the need for iterative traversal within KGs, which significantly reduces interactions between the LLM and the KG. (2) Our embedding-based extraction strategy avoids the locally optimal solution that arises from the stepwise interactions between LLMs and KGs, ensuring more effective exploration of the KGs. (3) KARPA operates in a training-free manner, making it adaptable

to various LLMs while enhancing the reasoning capabilities of LLMs over KGs through techniques such as chain-of-thought (CoT) (Wei et al., 2022). Our contributions can be summarized as follows:

- We propose KARPA, a framework that leverages the complementary strengths of LLMs and embedding models to improve both the accuracy and efficiency of KGQA tasks, while addressing the limitations of existing LLM-based methods.

- KARPA fully leverages the global planning and reasoning capabilities of LLMs in conjunction with a novel semantic embedding-based extraction method. In the pre-planning phase, the LLM is empowered to generate initial relation paths that are not restricted to adjacent relations, but can instead select from all potential relations within the KG, constructing logically coherent paths leading to answer entities. By integrating an embedding model to extract relation paths based on semantic similarity, KARPA mitigates the risk of the LLM getting trapped in local optima and significantly reduces the required interactions between the LLM and KGs. Techniques such as CoT prompting can also be incorporated to further enhance the LLM's reasoning abilities over KGs.

- Our KARPA framework operates in a training-free manner and can be seamlessly integrated with various LLMs, providing a plug-and-play solution that achieves state-of-the-art performance across multiple metrics on several KGQA benchmark datasets.

## 2 RELATED WORK

**Prompt-Based Reasoning with LLMs.**   Large Language Models (LLMs), such as LLaMA (Touvron et al., 2023a;b), Qwen (Bai et al., 2023), and GPT-4 (Achiam et al., 2023), have made substantial progress in enhancing reasoning capabilities by leveraging their vast internal knowledge. Various prompt-based methods have been proposed to further optimize these capabilities. For instance, Chain-of-Thought (CoT) prompting (Wei et al., 2022) facilitates a structured reasoning process by breaking down intricate tasks into manageable steps, significantly boosting performance in areas such as mathematical reasoning (Jie et al., 2023) and logical inference (Zhao et al., 2023). Building on CoT, several variants have been introduced to further optimize reasoning effectiveness, including Auto-CoT (Zhang et al., 2022), Zero-Shot-CoT (Kojima et al., 2022), and Complex-CoT (Fu et al., 2022). Additionally, newer frameworks like the Tree of Thoughts (ToT) (Yao et al., 2024) and Graph of Thoughts (GoT) (Besta et al., 2024) have expanded the scope of LLM reasoning, enabling the models to generate intermediate steps and sub-goals, thereby enhancing their versatility across diverse reasoning tasks. Lately, OpenAI o1 series models represent a significant advancement in LLM reasoning, allowing the LLM to develop an extensive internal chain of thought. These developments underscore the importance of tailored prompts in maximizing LLMs' reasoning potential.

**LLM-Based Knowledge Graph Question Answering.**   The integration of KGs with LLMs for question answering has emerged as a promising approach to enhance reasoning capabilities and mitigate hallucination phenomena. Unlike traditional CoT method that leverage the internal knowledge of LLMs, the incorporation of KGs facilitates access to structured external knowledge (He et al., 2022; Wang et al., 2023). Approaches such as Think-on-Graph (ToG) (Sun et al., 2023), Interactive-KBQA (Xiong et al., 2024) and StructGPT (Jiang et al., 2023) enable real-time interactions between LLMs and KGs. However, these methods often entail extensive interactions that can lead to inefficiencies. Reasoning on graphs (RoG) (Luo et al., 2023) uses instruction-tuned LLaMa2-Chat-7B to generate reasoning paths and achieves state-of-the-art performance on KGQA tasks. Similarly, methods such as chain of knowledge (Li et al., 2023c) and other approaches (Huang et al., 2024; Pan et al., 2024) employ LLMs to generate retrieval information for KGQA tasks. However, these methods require pre-training or fine-tuning process, which can be both costly and time-consuming. Additionally, methods such as UniKGQA (Jiang et al., 2022) and KG-CoT (Zhao et al., 2024) require the training of specific models for KG information retrieval, further complicating their implementation.

## 3 PRELIMINARY

In this section, we introduce key concepts and definitions relevant to our work, including Knowledge Graphs (KGs), relation paths, reasoning paths, Knowledge Graph Question Answering (KGQA), as well as embedding models and semantic similarity.

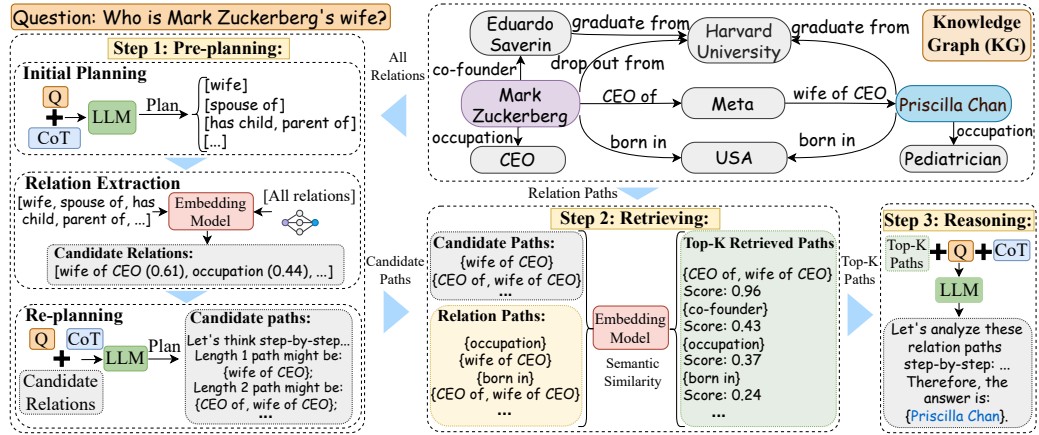

Figure 2: The framework of our KARPA. Our framework consists of three main steps: (1) Pre-planning: The LLM generates initial relation paths based on the given question. These paths are then decomposed for relation extraction using an embedding model. Utilizing the set of candidate relations, the LLM is able to re-plan logically coherent relation paths that potentially connect the topic entity and answer entities. (2) Retrieving: Candidate relation paths are extracted based on their similarity with re-planned initial paths, utilizing an embedding model. Our retrieval method accommodates paths that may differ in length from the re-planned initial paths. (3) Reasoning: The selected top-$K$ candidate relation paths are combined with the question and relevant entities to form a comprehensive prompt for the LLM, facilitating accurate question answering over the KG.

**Knowledge Graphs (KGs).** A Knowledge Graph (KG) is a structured representation of information, which can be represented as $G = (E, R)$, where $E$ denotes the set of entities and $R$ denotes the set of relations. Each relation $r \in R$ connects a pair of entities $(e_i, e_j)$ such that $e_i, e_j \in E$.

**Relation Paths and Reasoning Paths.** Relation paths are sequences of relations that connect two entities within a KG. A relation path $P$ from topic entity $e_t$ to answer entity $e_a$ can be expressed as: $P = (r_1, r_2, \ldots, r_n)$, where each $r_i \in R$ denotes the relations along the path. Reasoning paths extend this concept of relation paths by incorporating intermediate entities alone the path. A reasoning path $P_r$ from $e_t$ to $e_a$ can be represented as $P_r = \left\{ e_t \xrightarrow{r_1} e_1 \xrightarrow{r_2} e_2 \ldots \xrightarrow{r_n} e_a \right\}$.

**Knowledge Graph Question Answering (KGQA).** Knowledge Graph Question Answering (KGQA) involves the task of responding to questions by leveraging the information stored within KGs. Given a query $Q$, the goal of KGQA is to retrieve an answer $A$ defined as: $A = f(Q, G)$, where $f$ is a function that extracts the answer based on query $Q$ over the KG $G$.

**Embedding Models and Semantic Similarity.** Embedding Models facilitate the representation of words and sentences in a continuous vector space, enabling semantic embedding and similarity measurement. An embedding function $\Phi : R \to \mathbb{R}^d$ maps a sentence $R$ to $d$-dimensional vectors. The similarity between two embeddings can be quantified using metrics such as cosine similarity:

$$sim(r_i, r_j) = \frac{\Phi(r_i) \cdot \Phi(r_j)}{\|\Phi(r_i)\|\|\Phi(r_j)\|}, \tag{1}$$

where $\cdot$ denotes the dot product and $\|\cdot\|$ represents the Euclidean norm. This metric provides a measure of similarity between vectors, aiding in the retrieval and comparison of semantic information.

## 4 APPROACH

In this section, we present our proposed Knowledge graph Assisted Reasoning Path Aggregation (KARPA) framework, which leverages the strengths of LLMs and an embedding model to enhance KGQA. The approach consists of three key steps: pre-planning, retrieving, and reasoning.

## 4.1 PRE-PLANNING WITH LLM

The pre-planning phase is a crucial component of our KARPA framework, where we leverage the global planning capabilities of LLMs to generate initial relation paths $P_{initial}$. This phase initiates the reasoning process by allowing the LLM to analyze the input question $Q$ and the associated topic entity $e_t$. By leveraging the reasoning capability of LLM, KARPA is able to propose paths that are not only logically coherent but also have the potential to lead to the answer entities $E_a$.

**Initial Planning Using LLM**    KARPA start by leveraging the LLM's global planning capabilities to generate initial relation paths based on the provided question $Q$, as shown in Figure 2. The LLM outputs a set of potential relation paths $P$ as follows:

$$P = \{p_1, p_2, \ldots, p_m\} \quad \text{where } p_i = (r_1^i, r_2^i, \ldots, r_{n_i}^i) \text{ for } i = 1, 2, \ldots, m. \tag{2}$$

In Equation 2, each $p_i$ represents a relation path consisting of $n_i$ relations, $r_j^i \in R$, that are logically coherent and could connect a topic entity $e_t$ to potential answer entities $e_a$. The goal is to create several paths of varying lengths that could serve as candidates for relations extraction.

**Relation Extraction Strategy**    Once the initial relation paths $P$ are generated, we decompose each path $p_i$ into its constituent relations. For each path $p_i \in P$, the relations are organized into a relation list denoted as $R_i = \{r_1^i, r_2^i, \ldots, r_{n_i}^i\}$. For each relation $r_j^i$ in list $R_i$, we utilize an embedding model to extract top-$K$ semantically similar relations from the entire KG, as shown in Figure 2. This can be represented as:

$$R_j^i = \{r_{j1}, r_{j2}, \ldots, r_{jk}\} = \text{Top-K}(\text{sim}(\mathbf{r_j^i}, \mathbf{r})) \quad \text{for } r \in R, \tag{3}$$

where $\text{sim}(\cdot)$ denotes the semantic similarity function (e.g., cosine similarity) between the embedding of relation $r_j^i$ and all relations $r \in R$ using Equation 1. The resulting set $R_j^i$ contains the relations that best align semantically with the initial relations, ensuring that the LLM has access to relevant relations beyond just the immediate neighbors of current entity in the KG.

**Re-planning Relation Paths with LLM**    In the re-planning step, we leverage the candidate relations $R_j^i$ identified in the previous phase to construct formal relation paths that potentially connect the topic entity $e_t$ to the answer entity $e_a$. The process can be described as follows:

$$P_{initial} = \text{LLM}(Q, R_j^i), \text{ for each } r_j^i \in R_j^i \subset R. \tag{4}$$

Given the question $Q$ and candidate relations $R_j^i$, the LLM utilizes its global planning and reasoning capabilities to output initial relation paths $P_{initial}$, as shown in Figure 2. During this phase, we can integrate reasoning techniques like Chain-of-Thought (CoT) to further enhance the LLM's inference abilities on KGs. The CoT process encourages the LLM to consider the semantic connections between relations, leading to paths that are logically coherent.

By employing candidate relations extracted from the entire KG rather than being restricted to neighboring relations, our KARPA framework allows the LLM to construct the most logical reasoning chains without stepwise interactions between the LLM and KGs. This mitigates the risk of becoming trapped in local optima while reducing the required number of interactions. Through pre-planning process, we set the stage for effective retrieval and reasoning in the subsequent steps of our KARPA.

## 4.2 RELATION PATHS RETRIEVAL

In this section, we outline the retrieving step of our KARPA framework, which is designed to retrieve candidate relation paths in KGs. As shown in Figure 2, the retrieving process systematically explores potential relation paths derived from the initial paths generated by the LLM, providing candidate paths for reasoning step.

### 4.2.1 CONVENTIONAL RELATION PATHS RETRIEVAL

Conventional methods for LLM-based KG exploration ToG(Sun et al., 2023), typically involve the LLM selecting top-$K$ promising relations $R_t$ from the adjacent relations of the current entity $e$ at

each step. This strategy resembles a greedy algorithm, such as beam search. Formally, let $R(e)$ denote the set of relations available for the current entity $e$. The selection process can be defined as:

$$R_{\text{selected}} = \text{argmax}_{r \in R(e)} \, f(r), \; r \in KG. \tag{5}$$

In Equation 5, $f(r)$ is a scoring function indicating the potential of relation $r$. Since embedding similarity represents the similarity between two relations, we use $1 - sim(r_i, r_j)$ as the cost function for beam search. However, this approach does not guarantee finding the optimal path, as it may overlook globally optimal solutions.

To enhance relation path extraction, we employ traditional pathfinding algorithms like Dijkstra's, which can be expressed as:

$$cost(v) = \min\{cost(v), cost(v') + cost(v', v) \mid v' \text{ is a predecessor of } v\}. \tag{6}$$

In Equation 6, the cost to reach node $v$ is determined by either its current known cost or the cost of reaching one of its predecessors $v'$ plus $cost(v', v)$, the cost of the edge connecting $v'$ to $v$.

In KARPA, we begin from the topic entity $e_t$ and compute the semantic similarity $sim(r_i, r_j)$ using Equation 1 for relations at each step, scoring the relations based on their similarity to the corresponding relations in the initial relation paths $P_{initial}$. The cost for each step is defined as: $cost(r) = 1 - sim(r_i, r_j)$. This modification ensures that higher similarity scores correspond to lower costs, facilitating optimal path discovery. Since similarity scores range from 0 to 1, we average the total cost of relation paths of different lengths so that shorter paths can be fairly compared with longer paths. The path retrieval function based on Dijkstra's algorithm can be defined as:

$$cost(e) = \min \left\{ \frac{1}{n_e} cost(e), \frac{1}{n_{e'} + 1} \left[ cost(e') + sim(r_{(e',e)}, r_{initial}) \right] \right\}, \tag{7}$$

where the cost of entity $e$ is compared between $cost(e)$ averaged by the number of relations $n_e$ to reach entity $e$, and the cost of its predecessor $cost(e')$ plus the current cost $sim(r_{(e',e))}, r_{initial})$, averaged by number of relations $n_{e'}$ plus one. All current costs are computed between current relation and the corresponding relation in initial relation paths $P_{initial}$ using Equation 1.

### 4.2.2 HEURISTIC VALUE-BASED RELATION PATHS RETRIEVAL

Since the conventional relation paths retrieval methods require the cost of each relations alone the paths, the similarity between initial relation paths and current paths within the KG can only be calculated when current paths have the same length as initial paths $P_{initial}$. Inspired by the heuristic value in A* algorithm, we design a heuristic value-based relation paths retrieval method. In the traditional A* algorithm, the heuristic value serves as the a guiding function that indicates the distance between current node and target node. In KARPA, the heuristic value $h$ indicate the semantic similarity between the initial relation paths $P_{initial}$ and current path within the KG. By using heuristic value $h$ as an indicator, we are able to compute the similarity between paths of differing lengths, such as $A \xrightarrow{father} \xrightarrow{father} B$ and $A \xrightarrow{grandfather} B$, as shown in Figure 2. For paths $P_a$ and $P_b$, we concatenate all relations into one sentence and use the embedding model to calculate their similarity:

$$sim(P_a, P_b) = \frac{\text{emb}(\text{concat}(R(P_a))) \cdot \text{emb}(\text{concat}(R(P_b)))}{\|\text{emb}(\text{concat}(R(P_a)))\| \|\text{emb}(\text{concat}(R(P_b)))\|}. \tag{8}$$

In Equation 8, the similarity between path $P_a$ and $P_b$ can be calculated using the concatenation of their internal relations $R(P)$. Since the heuristic value represents the semantic distance between $P_a$ and $P_b$, it can be defined as $h = 1 - sim(P_a, P_b)$. The top-$K$ candidate relation paths $P_c$ with lowest heuristic value can be extracted as:

$$P_c = \text{argmax}_{P \in P_{all}} sim(P, P_{initial}), \; P_{all} \in KG. \tag{9}$$

Through Equation 9, we are able to identify and select the top-$K$ relevant paths from a diverse range of lengths as candidate paths $P_c$ for further reasoning.

The relation paths retrieval method in KARPA effectively broadens the search space and mitigates the risk of missing potentially optimal paths that traditional methods might overlook. The KARPA framework can dynamically adapt to various lengths of relation paths, even if the initial path of corresponding length does not exist. Through the retrieving step, we are able to extract the top-$K$ candidate relation paths for LLM to predict the finial answer for KGQA tasks.

| Type of Model | Method | WebQSP | | | CWQ | | |
|---|---|---|---|---|---|---|---|
| | | Accuracy | Hit@1 | F1 | Accuracy | Hit@1 | F1 |
| *Answering with Internal Knowledge* | | | | | | | |
| GPT-4 | IO prompt | - | 62.5 | - | - | 44.3 | - |
| GPT-4 | CoT* (Sun et al., 2023) | - | 67.3 | - | - | 46.0 | - |
| *Training-based Methods* | | | | | | | |
| LLaMA2-7B (Fine-tune) | KD-CoT* (Wang et al., 2023) | - | 68.6 | 52.5 | - | 55.7 | - |
| Graph Reasoning Model | KG-CoT* (Zhao et al., 2024) | - | ⬚84.9 | - | - | 62.3 | - |
| FiD-3B | DECAF* (Yu et al., 2022) | - | 82.1 | **78.8** | - | ⬚70.4 | - |
| PLM (Pretrain) | UniKGQA* (Jiang et al., 2022) | - | 77.2 | 72.2 | - | 51.2 | 49.0 |
| LLaMA2-7B (Fine-tune) | RoG | 80.4 | 84.6 | 70.1 | ⬚60.5 | 61.3 | ⬚54.2 |
| *Direct Inferance over KGs with LLMs* | | | | | | | |
| GPT-4o | ToG | 58.6 | 78.5 | 50.9 | 53.3 | 56.8 | 41.9 |
| GPT-4 | ToG* (Sun et al., 2023) | - | 82.6 | - | - | 69.5 | - |
| GPT-4o | KARPA | ⬚76.1 | 87.7 | 69.2 | 69.8 | 75.3 | 58.4 |
| GPT-4 | KARPA | **80.9** | **91.2** | ⬚72.1 | **73.6** | **78.4** | **61.5** |

Table 1: Comparison between our proposed KARPA and other baseline approaches. The table summarizes the performance of three categories of methods: (1) Answering with internal knowledge of LLMs, (2) Training-based methods, which require constant re-train for unseen KGs, and (3) Direct inference over KGs with LLMs. *Results are cited from corresponding publications. **Bold** represents the best result, underline represents the second best, and ⬚fbox represents the third best.

### 4.3 REASONING WITH LLM

In the reasoning step, we combine the candidate relation paths with their respective entities into a prompt for the LLM to reference during the final answer determination, as shown in Figure 2. The reasoning process of LLM can be formally expressed as:

$$Answer = \text{LLM}(Q, P_c, e_t, e_a), \ P_c = \{r_1, r_2, \dots, r_n\}. \tag{10}$$

Given the top-$K$ candidate relation paths $P_c$ and the question $Q$, the LLM can effectively assess whether the provided connections lead to a valid answer to $Q$. If the top-$K$ candidate paths do not yield a precise answer, we leverage the LLM's inherent knowledge to provide an appropriate response. The KARPA framework facilitates the LLM's ability to evaluate multiple reasoning paths in parallel, thereby enhancing the overall efficiency of LLM-based KGQA tasks.

## 5 EXPERIMENTS

In this section, we detail the experimental setup, present our main results, and conduct further analysis to evaluate the performance of our proposed Knowledge graph Assisted Reasoning Path Aggregation (KARPA) framework.

### 5.1 EXPERIMENTAL SETTINGS

**Datasets and Evaluation Metrics** We evaluate KARPA on two widely used multi-hop KGQA datasets: WebQuestionSP (WebQSP) (Yih et al., 2016) and Complex WebQuestions (CWQ) (Talmor, 2018). These two datasets are designed for Multi-hop KGQA tasks. We compare our proposed KARPA and other LLM-based KGQA methods to demonstrate the effectiveness of our framework. For evaluation, we employ three metrics: Accuracy, Hit@1, and F1 score. Accuracy measures the proportion of correctly answered questions. Hit@1 evaluates whether the correct answer is among the top predicted answers. F1 score combines precision and recall into a single metric, offering a balance evaluation between the two metrics.

**Baselines for Comparison** We compare KARPA against several baselines: (1) To demonstrate that KARPA derives answers through KG reasoning rather than relying on the internal knowledge of the LLM, we report the result of IO Prompt (Brown et al., 2020), which directly answers questions without a reasoning process. The result of CoT (Wei et al., 2022) is also included as a baseline

| Model Tpye | Method | WebQSP | | | CWQ | | |
|---|---|---|---|---|---|---|---|
| | | Accuracy | Hit@1 | F1 | Accuracy | Hit@1 | F1 |
| GPT-4o-mini | CoT | - | 61.3 | - | - | 49.5 | - |
| | ToG | 56.4 | 75.2 | 51.6 | 50.2 | 54.0 | 34.5 |
| | KARPA | **71.9** | **85.3** | **64.5** | **68.1** | **73.3** | **56.5** |
| GPT-4o | CoT | - | 67.0 | - | - | 52.3 | - |
| | ToG | 58.6 | 78.5 | 50.9 | 53.3 | 56.8 | 41.9 |
| | KARPA | **76.1** | **87.7** | **69.2** | **69.8** | **75.3** | **58.4** |
| GPT-4 | CoT | - | 66.1 | - | - | 54.7 | - |
| | ToG* | - | 82.6 | - | - | 69.5 | - |
| | KARPA | **80.9** | **91.2** | **72.1** | **73.6** | **78.4** | **61.5** |
| Claude-3.5-Sonnet | CoT | - | 72.3 | - | - | 57.4 | - |
| | ToG | 61.5 | 79.2 | 53.4 | 54.1 | 60.3 | 43.5 |
| | KARPA | **82.6** | **89.5** | **69.7** | **70.7** | **73.6** | **54.9** |
| Gemini-1.5-Pro | CoT | - | 65.3 | - | - | 52.1 | - |
| | ToG | 62.3 | 78.4 | 52.5 | 51.7 | 57.9 | 40.5 |
| | KARPA | **80.7** | **90.5** | **68.6** | **69.8** | **75.0** | **54.8** |

Table 2: Comparison of our proposed KARPA, ToG, and CoT using various LLMs. The results demonstrate that KARPA consistently outperforms ToG, the previous state-of-the-art for direct KG-based reasoning using LLM. *Results of ToG are cited from corresponding paper (Sun et al., 2023).

to evaluate the LLM's reasoning performance without external knowledge. (2) KARPA is further compared with training-based KGQA methods, including KD-CoT (Wang et al., 2023), UniKGQA (Jiang et al., 2022), DECAF (Yu et al., 2022), and RoG (Luo et al., 2023). This comparison demonstrates that KARPA effectively leverages the LLM's planning and reasoning capabilities without additional training. (3) Lastly, KARPA is compared with ToG (Sun et al., 2023), the current state-of-the-art method that operates without training.

**Experimental Details** We test various LLMs including GPT-4 (OpenAI, 2023), GPT-4o (OpenAI, 2024), GPT-4-mini, Claude-3.5-Sonnet (Anthropic, 2024), Gemini-1.5-pro (Team et al., 2024) and other models via API calls. We employ all-MiniLM-L6-v2 based on sentence-transformers (Reimers, 2019) as the embedding model. For each LLM, we randomly select 300 KGs from each datasets (WebQSP, CWQ) to evaluate KARPA's performance, aiming to reduce computational costs.

In implementing KARPA, we determine that the initial relation paths planned by the LLM during pre-planning step represent the most reasonable path lengths. Therefore, during the retrieving step, we only extract paths that match the length of the initial paths predicted by the LLM. In the retrieving step based on beam search and pathfinding algorithms, we set the number of top-$K$ paths to 16, selecting 16 paths with the highest semantic similarity for each initial relation path as candidate paths. In the heuristic value-based retrieval step, since our method can compute the similarity between paths of different lengths, we select 16 paths with the highest similarity for each initial path from relation paths of various lengths, which are then used as candidate paths for the reasoning step.

## 5.2 MAIN RESULTS

### 5.2.1 COMPARISON BETWEEN BASELINES

We evaluate our method against the following approaches: direct answering with GPT-4 (IO prompt), reasoning with internal knowledge (CoT), training-based methods and direct interaction with KGs (ToG). We present the results in Table 1. The results show that our method significantly outperforms existing approaches across most metrics, achieving state-of-the-art performance. When comparing our framework to the direct answering with internal knowledge, we demonstrate that leveraging KGs as external knowledge sources enables the LLM to yield superior answers.

In contrast to training-based methods, our approach offers the advantage of being plug-and-play, requiring no additional training while still ensuring effective reasoning based on the KGs. Furthermore, our results indicate that KARPA generalizes well across different KGQA datasets. When comparing with the ToG method, which also utilizes LLMs for reasoning over KGs without ad-

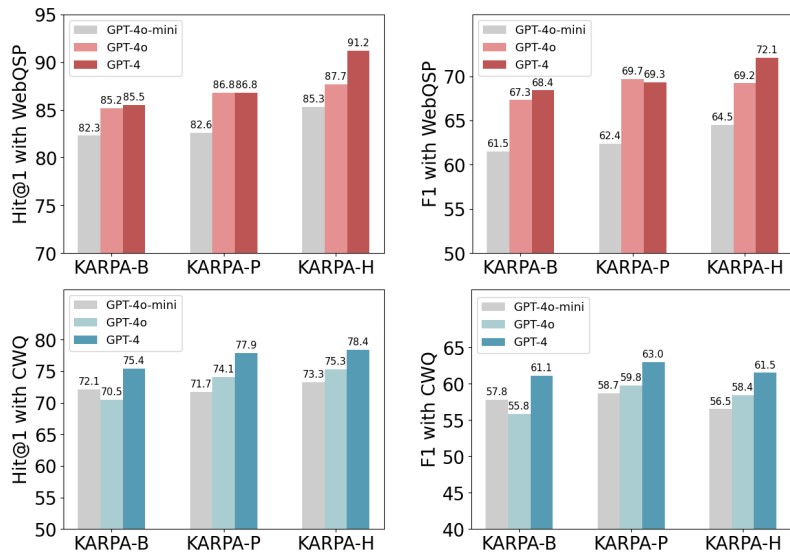

Figure 3: Comparison of different retrieval strategies across various LLMs on Hit@1 and F1 metrics. Results illustrate the performance of KARPA-B (beam search-based), KARPA-P (pathfinding-based), and KARPA-H (heuristic value-based) retrieval strategies when using different LLMs.

ditional training (Sun et al., 2023), our KARPA framework achieves notably better results across all metrics. This underscores the value of integrating global planning capabilities with the LLM's reasoning process, allowing for the construction of logically coherent relation paths that effectively direct the LLM from the topic entity to the answer entities.

### 5.2.2 PERFORMANCE ACROSS DIFFERENT LLMS

We also evaluate ToG and KARPA with different LLMs, including GPT-4, GPT-4o, GPT-4-mini, Claude-3.5 - Sonnet, and Gemini-1.5-pro. Both ToG and our KARPA approach rely on the reasoning capabilities of these LLMs without requiring additional training. The results, shown in Table 2, indicate that KARPA consistently outperforms ToG, regardless of the LLM used. This demonstrates that KARPA's ability to harness LLMs' global planning and reasoning capabilities allows it to construct more logically sound and complete reasoning chains, which ultimately lead to more accurate answers. In contrast, ToG's reliance on stepwise relation selection limits its effectiveness, as it neglects the LLM's inherent planning capabilities.

| Method | WebQSP | CWQ |
|---|---|---|
| ToG* | 11.2 | 14.3 |
| KARPA+GPT-4o-mini | 5.1 | 6.2 |
| KARPA+GPT-4o | **4.8** | **5.3** |
| KARPA+GPT-4 | 5.5 | 6.0 |
| KARPA+Claude | 6.6 | 7.3 |
| KARPA+Gemini | 5.8 | 7.4 |

Table 3: Comparison of LLM call frequency. The LLM call of ToG are cited from its paper.

Additionally, we evaluate the performance of these LLMs when using CoT prompting. Our results clearly show that when KG information is incorporated, the LLMs are able to provide more accurate and complete answers, further emphasizing the value of external knowledge sources like KGs in enhancing LLM reasoning capabilities.

### 5.3 FURTHER ANALYSIS

In this section, we conduct a deeper analysis of KARPA, exploring two key aspects: (a) the comparison of interaction steps between KARPA and the baseline method ToG, and (b) ablation studies to evaluate the impact of different retrieval methods and LLMs on the performance of KARPA.

### 5.3.1 INTERACTION STEPS COMPARISON

We evaluate the average number of interactions required to obtain an answer for both ToG and KARPA across multiple LLMs and datasets. The results, presented in Table 3, show that KARPA

consistently reduces the number of interactions by more than half compared to ToG, while maintaining superior performance in terms of answer accuracy and reasoning quality.

The primary reason for this efficiency lies in the differences between the interaction mechanisms of the two approaches. In ToG, the stepwise relation selection on KGs is not only time-consuming but also leads to a higher demand for computational resources during interaction with the KG. In contrast, KARPA requires only two interactions with the LLM during the pre-planning step to generate the initial relation paths. These initial paths form a coherent reasoning chain that serves as the backbone for the subsequent retrieval process. Instead of repeatedly invoking the LLM for relation extracting, KARPA leverages an embedding model to extract similar relation paths from the KG based on semantic similarity. This significantly reduces the overall interaction steps and the computational cost of KG-based reasoning.

### 5.3.2 ABLATION STUDIES

We perform two sets of ablation studies to further understand the components of our approach and how they contribute to its effectiveness.

**Impact of different retrieval methods.** In the retrieving phase of KARPA, we experiment with different methods to extract relation paths and analyze their impact on the final results. The comparison is shown in Table 4, where we evaluate three retrieval strategies: (1) **KARPA-B**: A beam search-based retrieval method with a fixed beam width to extract relation paths. This method is similar to ToG in that it calculates semantic similarity for paths using stepwise interactions. (2) **KARPA-P**: A pathfinding-based retrieval method that calculates the semantic similarity between relation paths based on pre-defined distance metrics, constrained to extracting paths of the same length as the initial relation paths. (3) **KARPA-H**: A heuristic value-based retrieval method that is able to compute semantic similarity between paths of different lengths, allowing more flexibility in the candidate path selection process.

| Method | WebQSP | CWQ |
|---|---|---|
| GPT-4o-mini | Hit@1 | Hit@1 |
| KARPA-B | 82.3 | 72.1 |
| KARPA-P | 82.6 | 71.8 |
| KARPA-H | **85.3** | **73.3** |
| GPT-4o | Hit@1 | Hit@1 |
| KARPA-B | 85.2 | 70.5 |
| KARPA-P | 86.8 | 74.0 |
| KARPA-H | **87.7** | **75.3** |

Table 4: Hit@1 value of KARPA with various retrieval strategies.

The results indicate that KARPA-H outperforms other retrieval methods, providing superior KGQA results when using the same LLMs. Additional results are provided in Appendix C.

**Influence of different LLMs.** We also examine how different LLMs affect the performance of our method, as shown in Figure 3. Since KARPA relies on the global planning and reasoning capabilities of LLMs, the strength of the LLM plays a significant role in the overall performance of the KARPA.

The results indicate that more powerful LLMs (such as GPT-4) generate better initial paths, leading to more accurate question answering (Kaplan et al., 2020). Conversely, when using the weaker LLM (e.g., GPT-4o-mini), the performance of KARPA slightly declines, though it still outperforms the ToG method. This demonstrates the importance of strong reasoning capabilities in the LLMs for KG-based tasks. The findings also suggest that LLMs with better planning and reasoning abilities can extract more meaningful insights from KGs, thus enhancing overall accuracy of KGQA tasks.

## 6 CONCLUSION

In this paper, we propose KARPA, a novel framework designed to enhance LLM-based KGQA by utilizing the global planning and reasoning capabilities of LLMs. KARPA addresses key limitations of existing approaches by improving both accuracy and efficiency, while providing a plug-and-play solution through its structured pre-planning, retrieving, and reasoning processes. Our experiments demonstrate that KARPA consistently outperforms state-of-the-art methods across multiple datasets and evaluation metrics. Furthermore, its training-free nature enables seamless integration with a variety of LLMs, offering broad applicability to different KGQA tasks. By optimizing LLM-KG interactions, KARPA improves reasoning efficiency and effectiveness, highlighting its potential as a robust approach for future retrieval-augmented generation (RAG) systems.

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

## A    ALGORITHM FOR KARPA

In this section, we present the pseudo-code for the Knowledge graph Assisted Reasoning Path Aggregation (KARPA) framework, as shown in Algorithm 1. The pseudo-code outlines the key components of our approach, including the pre-planning, retrieval, and reasoning phases. It demonstrates the interaction between the large language model (LLM) and the embedding model in generating, retrieving, and refining relation paths, which are crucial for improving LLM-based KGQA tasks.

---

**Algorithm 1:** KARPA Framework

**Input:** Question $Q$, Topic entity $e_t$, Knowledge Graph $KG$, Large Language Model $LLM$, Embedding Model

**Output:** Answers $E_a$

**Pre-Planning Phase:**

Generate initial paths $P_i = \{p_1, p_2, \ldots, p_m\}$ using $LLM(Q, e_t)$;

**for** *each path $p_i = (r_1^i, r_2^i, \ldots, r_{n_i}^i)$* **do**

    Decompose $p_i$ into relation list $R_i = \{r_1^i, r_2^i, \ldots, r_{n_i}^i\}$;

    **for** *each relation $r_j^i$ in $R_i$* **do**

        Retrieve top-$K$ similar relations $R_j^i = \text{Top-K}(\text{sim}(\mathbf{r_j^i}, \mathbf{r}))$;

    **end**

**end**

Re-plan relation paths $P_{replan} = \text{LLM}(Q, R_j^i)$ based on retrieved relations $R_j^i$;

**Retrieving Phase:**

Extract relation paths $P_r$ with length $L \in len(P_{replan})$;

**for** *each path $p$ in $P_{replan}$* **do**

    Compute similarity between paths using heuristic value

    $P_{retrieved} = \text{Heuristic}(\text{sim}(p, p_r), p_r \in P_r)$;

    Retrieve top-$K$ similar paths $P = \text{Top-K}(P_{retrieved})$ as $P_{candidate}$;

**end**

**Reasoning Phase:**

Combine candidate relation paths $P_{candidate} = \{r_1, r_2, \ldots, r_n\}$ with $e_t, e_a$ into prompt;

Predict final answer $E_a = \text{LLM}(Q, P_{candidate}, e_t, e_a)$;

**return** $E_a$

---

## B    IMPLEMENTATION DETAILS

**Model Invocation.**    Our method, KARPA, along with the baseline comparison methods such as CoT (Wei et al., 2022) and ToG (Sun et al., 2023), is all implemented via API calls to various large language models (LLMs). These LLMs are queried dynamically throughout the experimental pipeline to perform pre-planning, retrieving, and reasoning steps.

**Experimental Setup.**    During the pre-planning stage, the initial paths generated by the LLM are decomposed and stored, along with the query, into a list. For each element in this list, we retrieve the top-k relations, where the total number of retrieved relations does not exceed 30. These relations are semantically closest to the elements based on the LLM's initial output.

In the retrieving step, KARPA selects the top 16 relation paths with the highest similarity for each initial relation path. These paths serve as candidate paths for reasoning step. In the reasoning step, we limit the number of candidate paths input to the LLM at one time to a maximum of 8, ensuring that the reasoning process remains manageable and focused on the most relevant paths.

| Model Tpye | Method | WebQSP | | | |
|---|---|---|---|---|---|
| | | Accuracy | Hit@1 | F1 | Precision |
| GPT-4o-mini | KARPA-B | 67.2 | 82.3 | 61.5 | 64.1 |
| | KARPA-P | 67.8 | 82.6 | 62.4 | 64.9 |
| | KARPA-H | **71.9** | **85.3** | **64.5** | **65.9** |
| GPT-4o | KARPA-B | 73.8 | 85.2 | 67.3 | 72.3 |
| | KARPA-P | 73.7 | 86.8 | **69.7** | 70.5 |
| | KARPA-H | **76.1** | **87.7** | 69.2 | **71.5** |
| GPT-4 | KARPA-B | 73.5 | 85.5 | 68.4 | 71.7 |
| | KARPA-P | 74.1 | 86.8 | 69.3 | **73.6** |
| | KARPA-H | **80.9** | **91.2** | **72.1** | 73.1 |
| DeepSeek-V2.5 | KARPA-B | 71.8 | 84.0 | 63.1 | 65.9 |
| | KARPA-P | 73.4 | 85.3 | 64.1 | 66.3 |
| | KARPA-H | **78.1** | **88.4** | **68.7** | **67.6** |
| Gemini-1.5-Pro | KARPA-B | 70.1 | 84.5 | 65.9 | 64.7 |
| | KARPA-P | 73.8 | 88.0 | 67.4 | 66.1 |
| | KARPA-H | **80.7** | **90.5** | **68.6** | **67.8** |
| Claude-3.5-Sonnet | KARPA-B | 75.1 | 85.7 | 66.0 | 67.6 |
| | KARPA-P | 80.4 | 89.0 | 69.7 | **70.4** |
| | KARPA-H | **82.6** | **89.5** | **69.7** | 69.1 |

Table 5: Performance of KARPA with different retrieval strategies (KARPA-B, KARPA-P, and KARPA-H) and LLMs on the WebQSP dataset.

**Answer Evaluation.** To determine if the LLM correctly answers the question, KARPA enforces a specific output format. The final answer must be enclosed in curly brackets in the LLM's output. We consider an answer correct only when the tail entities of the reasoning paths match the text enclosed within the curly brackets in the LLM's output. For CoT, we consider an answer correct if the LLM's response contains the correct answer entities. This difference reflects the distinct reasoning and output expectations between KARPA and CoT.

## C ADDITIONAL RESULTS

In this section, we present additional experimental results to further evaluate the performance of KARPA when using different retrieval methods: KARPA-B (beam search-based retrieval), KARPA-P (pathfinding-based retrieval), and KARPA-H (heuristic value-based retrieval). We conduct these experiments across various LLMs, analyzing the effectiveness of each retrieval strategy in conjunction with different LLMs. These results provide a deeper insight into how different retrieval mechanisms impact the overall performance of KARPA, showcasing the versatility and adaptability of our approach under varying model conditions.

The results presented in Table 5 and Table 6 consistently demonstrate the superior performance of KARPA-H (heuristic value-based retrieval) compared to the other two retrieval strategies, KARPA-B (beam search-based) and KARPA-P (pathfinding-based), across different LLMs and datasets (WebQSP and CWQ).

In the majority of LLMs, KARPA-H outperforms the other methods in most metrics. This suggests that KARPA-H is more effective at extracting the correct relation paths, which in turn leads to more accurate and contextually relevant answers. These results highlight KARPA-H as the most robust and reliable retrieval method among the three, reinforcing its advantage in handling complex KG-based reasoning tasks.

## D ADDITIONAL EXPERIMENTS

| Model Tpye | Method | CWQ | | | |
|---|---|---|---|---|---|
| | | Accuracy | Hit@1 | F1 | Precision |
| GPT-4o-mini | KARPA-B | 66.0 | 72.1 | 57.8 | 58.6 |
| | KARPA-P | 66.4 | 71.7 | **58.7** | **59.8** |
| | KARPA-H | **68.1** | **73.3** | 56.5 | 55.1 |
| GPT-4o | KARPA-B | 65.0 | 70.5 | 55.8 | 57.8 |
| | KARPA-P | 69.2 | 74.1 | **59.8** | 58.4 |
| | KARPA-H | **69.8** | **75.3** | 58.4 | **59.5** |
| GPT-4 | KARPA-B | 71.2 | 75.4 | 61.1 | 62.7 |
| | KARPA-P | 73.4 | 77.9 | **63.0** | 62.5 |
| | KARPA-H | **73.6** | **78.4** | 61.5 | **63.1** |
| DeepSeek-V2.5 | KARPA-B | 61.6 | 63.2 | 48.4 | 50.1 |
| | KARPA-P | 60.9 | 63.0 | 51.8 | 52.6 |
| | KARPA-H | **62.6** | **64.1** | **51.9** | **53.5** |
| Gemini-1.5-Pro | KARPA-B | 69.1 | 74.0 | 57.2 | 59.5 |
| | KARPA-P | 69.6 | 73.5 | **57.7** | **60.3** |
| | KARPA-H | **69.8** | **75.0** | 54.8 | 55.8 |
| Claude-3.5-Sonnet | KARPA-B | 62.8 | 65.7 | 49.6 | 52.1 |
| | KARPA-P | 61.5 | 64.3 | 52.9 | 55.5 |
| | KARPA-H | **70.6** | **73.7** | **54.9** | **56.9** |

Table 6: Performance of KARPA with different retrieval strategies (KARPA-B, KARPA-P, and KARPA-H) and LLMs on the CWQ dataset.

In this section, we provide additional experiments to validate KARPA's performance from different perspectives.

To demonstrate that KARPA has better generalization capabilities than methods based on instruction-tuned LLMs, we conducted an experiment using GPT-4o-mini with a modified version of the WebQSP dataset. Specifically, we slightly alter the questions in WebQSP dataset while preserving their original meaning, using the prompt: "Please revise the question to make it more clear, but the original meaning of the question and the corresponding answers remain unchanged." We test RoG using its instruction-tuned LLaMa2-Chat-7B from in the planning step and GPT-4o-mini for reasoning. In KARPA, we use GPT-4o-mini for both pre-planning and reasoning steps.

| Question | Method | Accuracy | Hit@1 | F1 | Method | Accuracy | Hit@1 | F1 |
|---|---|---|---|---|---|---|---|---|
| Origin | RoG | 67.6 | 84.1 | 69.7 | KARPA | 73.1 | 85.4 | 68.1 |
| Revised | RoG | 63.5 | 74.3 | 64.1 | KARPA | 72.6 | 84.5 | 68.9 |
| Variation | RoG | -4.1 | -9.8 | -5.6 | KARPA | -0.5 | -0.9 | +0.8 |

Table 7: Comparison of RoG and KARPA on the WebQSP dataset with original and revised questions.

The results in Table 7 show that KARPA's performance remains consistent and robust to question modifications, while RoG's performance drops due to path mismatches. This further highlights the advantage of KARPA's training-free framework, maintaining superior robustness and adaptability across all KGs.

We also conduct an additional experiment using instruction-tuned LLaMa2-Chat-7B as the backbone LLM for both KARPA and RoG, while using untrained Qwen2.5-7B and Qwen2.5-14B for final answer reasoning in both methods.

The results in Table 8 show that with the same backbone LLM, KARPA's semantic similarity-based retrieval methods successfully extract more accurate reasoning paths, leading to higher accuracy in final answers.

| Base-model | Method | WebQSP | | | CWQ | | |
|---|---|---|---|---|---|---|---|
| | | Accuracy | Hit@1 | F1 | Accuracy | Hit@1 | F1 |
| LLaMa2-7B + Qwen2.5-7B | RoG | 54.5 | 73.8 | 57.2 | 38.6 | 43.5 | 35.8 |
| | KARPA | 66.4 | 82.7 | 63.6 | 54.1 | 59.2 | 46.3 |
| LLaMa2-7B + Qwen2.5-14B | RoG | 58.7 | 77.2 | 60.9 | 43.9 | 48.0 | 42.5 |
| | KARPA | **69.8** | **84.2** | **67.4** | **55.0** | **60.4** | **47.2** |

Table 8: Comparison of RoG and KARPA performance on WebQSP and CWQ datasets using instruction-tuned LLaMa2-Chat-7B as the backbone LLM.

We also compare KARPA with Interactive-KBQA (Xiong et al., 2024), a robust agent-like method which directly perform inference over KGs with LLMs. Interactive-KBQA shares similarities with ToG as both approaches rely on direct, step-by-step interaction between LLMs and KGs to infer answers. In contrast, KARPA eliminates the need for iterative interaction by directly generating a complete reasoning path based on relations extracted from the KG. Our approach significantly reduces the computational cost for LLMs and improves the logical coherence of reasoning paths. To further substantiate KARPA's advantages, we conduct an additional experiment comparing KARPA with Interactive-KBQA, using GPT-4-turbo as the backbone LLM. The results of Interactive-KBQA are cited from its paper.

| Method | 1-hop | 2-hop | Overall | RHits@1 | Overall (CWQ) |
|---|---|---|---|---|---|
| Interactive-KBQA | 69.99 | 72.41 | 71.20 | 72.47 | 49.07 |
| KARPA | **74.21** | **72.97** | **73.78** | **74.14** | **61.45** |

Table 9: Comparison of Interactive-KBQA and KARPA performance on WebQSP and CWQ datasets.

In Table 9, 1-hop and 2-hop represent the F1 scores on the WebQSP dataset for KG with reasoning paths of length 1 and length 2, respectively. Overall refers to the overall F1 score on the WebQSP dataset. Random Hit@1 (RHit@1) is calculated following the method used in TIARA (Shu et al., 2022), where an answer is randomly selected for each question 100 times, and the average Hits@1 is reported. Overall (CWQ) represents the overall F1 score on the CWQ dataset. The results show that KARPA outperforms Interactive-KBQA on WebQSP and CWQ datasets with GPT-4-turbo.

To demonstrate the impact of different embedding models on KARPA, we conduct additional experiments comparing various embedding models to evaluate their effects on KARPA's performance when using GPT-4o-mini.

| Embedding Model | WebQSP | | | CWQ | | |
|---|---|---|---|---|---|---|
| | Accuracy | Hit@1 | F1 | Accuracy | Hit@1 | F1 |
| all-MiniLM-L6-v2 | 72.3 | **86.4** | 67.2 | 64.6 | 67.7 | 55.1 |
| all-mpnet-base-v2 | **74.5** | 86.1 | **68.6** | 64.1 | 68.3 | 53.7 |
| multilingual-MiniLM-L12-v2 | 74.1 | 85.3 | 68.3 | **65.3** | **69.5** | **55.4** |

Table 10: Performance comparison of different embedding models on WebQSP and CWQ datasets.

In Table 10, all-MiniLM-L6-v2 is the default embedding model used in KARPA, with a size of approximately 86MB. all-mpnet-base-v2, a more powerful embedding model, is around 417MB. paraphrase-multilingual-MiniLM-L12-v2, which supports embedding between multiple languages, has a size of approximately 448MB. The results demonstrate that KARPA's robust design ensures that its overall performance remains consistent across different embedding models. This is because the candidate paths generated by KARPA during the pre-planning phase are very distinct. While they are semantically close to the correct reasoning paths, they differ significantly from incorrect reasoning paths. Therefore, a basic embedding model is sufficient to assist KARPA in extracting the correct paths.

We also provide the Exact Match (EM) metric (Talmor & Berant, 2018) for a more comprehensive analysis. The results in Table 11 demonstrate that KARPA achieves higher EM scores compared to ToG, showing its effectiveness in accurately extracting reasoning paths and final answers.

| Base-Model | Method | EM (WebQSP) | EM (CWQ) |
|---|---|---|---|
| GPT-4o | ToG | 39.5 | 37.6 |
| GPT-4o | KARPA | 44.6 | 41.3 |
| GPT-4 | ToG | 43.1 | 40.9 |
| GPT-4 | KARPA | **51.7** | **47.2** |

Table 11: Exact Match (EM) performance comparison between ToG and KARPA on WebQSP and CWQ datasets.

To demonstrate the effectiveness of KARPA with smaller LLMs, we conduct additional experiments with Qwen2.5-7B and Qwen2.5-14B as the LLM backbones for KARPA. The results in Table 12 demonstrate that KARPA consistently outperforms stepwise direct inference baselines such as ToG, even when using smaller LLMs. This reinforces the robustness and adaptability of our method across different LLM scales.

| Base-Model | Method | WebQSP | | | CWQ | | |
|---|---|---|---|---|---|---|---|
| | | Accuracy | Hit@1 | F1 | Accuracy | Hit@1 | F1 |
| Qwen2.5-7B | CoT | - | 41.5 | - | - | 28.3 | - |
| | ToG | 24.6 | 30.2 | 21.9 | 22.4 | 25.8 | 20.2 |
| | KARPA | 65.6 | 79.2 | 58.6 | 47.6 | 52.7 | 38.8 |
| Qwen2.5-14B | CoT | - | 49.6 | - | - | 31.2 | - |
| | ToG | 45.0 | 55.9 | 42.7 | 31.2 | 36.6 | 29.5 |
| | KARPA | **72.6** | **84.1** | **65.0** | **51.5** | **57.9** | **41.6** |

Table 12: Performance comparison of different methods on WebQSP and CWQ datasets using smaller LLMs.

Also, the results in Table 12 show that KARPA can perform well with LLMs that have weaker planning and reasoning capabilities, further highlighting KARPA's robustness and its reduced dependence on the LLM's planning and reasoning abilities compared to other inference-based methods.

To quantify the impact of the re-planning step, we provide an ablation study that removes the re-planning step from the pre-planning stage. The re-planning step is designed to handle mismatches between LLMs and KGs. In re-planning step, the extracted relations are used to refine and re-plan candidate paths. This guarantees that the candidate paths are both logically coherent and aligned with the KG.

| Pre-Planning | WebQSP | | | CWQ | | |
|---|---|---|---|---|---|---|
| | Accuracy | Hit@1 | F1 | Accuracy | Hit@1 | F1 |
| Origin | 72.3 | 86.4 | 67.2 | 64.6 | 67.7 | 55.1 |
| Remove Re-Planning Step | 64.1 | 79.6 | 61.5 | 54.3 | 59.5 | 47.1 |

Table 13: Ablation study of removing re-planning step from the pre-planning stage.

The results in Table 13 show that the re-planning step is crucial for KARPA's performance. Additionally, in the retrieval step, KARPA employs semantic similarity as the cost function for pathfinding algorithms. This ensures that the final reasoning paths selected not only exist in the KG but are also semantically closest to the paths generated by the LLM, thereby maintaining the validity of the LLM's output across diverse query problems.

To demonstrate that KARPA reduces the logical complexity of LLM reasoning on KGs, we provide a comparison of the average number of input and output tokens between ToG and KARPA using the

tokenizer of GPT-4o-mini. Methods that rely on step-by-step interactions between the LLM and KG must select the next relations from hundreds or even thousands of adjacent relations at each step, and repeat this process until the answer entities are found. This results in a high computational burden, and also fails to leverage the LLM's global planning capabilities.

| Method | WebQSP | | CWQ | |
| --- | --- | --- | --- | --- |
| | Input Tokens/KG | Output Tokens/KG | Input Tokens/KG | Output Tokens/KG |
| ToG | 6351.5 | 1836.5 | 7935.7 | 2931.6 |
| KARPA | 2465.9 | 1492.3 | 3612.1 | 2267.1 |

Table 14: Token usage comparison between ToG and KARPA on WebQSP and CWQ datasets.

The results in Table 14 show that KARPA significantly reduces both input and output token usage compared to ToG, which means we have not only lowered the reasoning complexity for the LLM but also saved on the computational costs of the LLM, further demonstrating the superiority of KARPA.

The multilingual scenarios can be effectively addressed by using multilingual embedding models. For instance, in a multilingual setting, we test KARPA with paraphrase-multilingual-MiniLM-L12-v2, a multilingual embedding model. In the multilingual experiment, we use GPT-4o-mini to generate relation paths in Chinese, and then use the multilingual embedding model to calculate the semantic similarity between the candidate paths and paths in the KG.

| Language | WebQSP | | | CWQ | | |
| --- | --- | --- | --- | --- | --- | --- |
| | Accuracy | Hit@1 | F1 | Accuracy | Hit@1 | F1 |
| English-English | 74.1 | 85.3 | 68.3 | 65.3 | 69.5 | 55.4 |
| Chinese-English | 74.6 | 84.5 | 67.6 | 63.1 | 68.0 | 54.2 |

Table 15: Performance comparison of different languages using a multilingual embedding model.

These results in Table 15 demonstrate that with a multilingual embedding model, KARPA performs effectively across languages, maintaining its robustness. They also indicate that language variations do not significantly impact KARPA's performance.

To demonstrate the necessity of extending relation paths with different lengths, we restrict the retrieval step to use only single-relation candidate paths provided by the LLM during re-planning step, and compare the performance of the heuristic value-based retrieval method (KARPA-H) with the pathfinding-based retrieval method (KARPA-P) using GPT-4o-mini.

| Candidate Path | Method | WebQSP | | | CWQ | | |
| --- | --- | --- | --- | --- | --- | --- | --- |
| | | Accuracy | Hit@1 | F1 | Accuracy | Hit@1 | F1 |
| Original Paths | KARPA-P | 66.0 | 81.2 | 63.8 | 61.0 | 64.5 | 53.4 |
| Original Paths | KARPA-H | 72.3 | 86.4 | 67.2 | 64.6 | 67.7 | 55.1 |
| Single-Relation Paths | KARPA-P | 63.6 | 77.3 | 60.7 | 40.5 | 43.9 | 39.3 |
| Single-Relation Paths | KARPA-H | 71.4 | 85.5 | 68.9 | 55.1 | 59.6 | 47.4 |

Table 16: Performance of KARPA-P and KARPA-H using different candidate paths on the WebQSP and CWQ datasets.

The results in the Table 16 demonstrate that the heuristic value-based retrieval method outperforms pathfinding-based retrieval methods in such scenarios, as it effectively addresses the semantic similarity issues that arise from differing path lengths. Moreover, as the questions in the CWQ dataset generally require longer reasoning paths compared to WebQSP, both methods exhibit a more significant decline in various metrics on CWQ. However, the heuristic value-based retrieval method shows a less pronounced drop compared to pathfinding-based retrieval methods, further demonstrating its superiority.

To validate the performance of KARPA on KGs outside the training scope, we compare KARPA with Chain-of-Thought (CoT) reasoning, where the LLM directly relies on its internal knowledge to an-

swer questions. Using smaller-scale LLMs such as Qwen2.5-7B, Qwen2.5-14B and Qwen2.5-72B (with limited stored knowledge), we observe that CoT performance drops significantly on KGQA tasks while KARPA maintains strong performance.

| Base-Model | Method | WebQSP | | | CWQ | | |
|---|---|---|---|---|---|---|---|
| | | Accuracy | Hit@1 | F1 | Accuracy | Hit@1 | F1 |
| | CoT | - | 41.5 | - | - | 28.3 | - |
| Qwen2.5-7B | KARPA | 65.6 | 79.2 | 58.6 | 47.6 | 52.7 | 38.8 |
| | **Gain** | - | **+37.7** | - | - | **+24.4** | - |
| | CoT | - | 49.6 | - | - | 31.2 | - |
| Qwen2.5-14B | KARPA | 72.6 | 84.1 | 65.0 | 51.5 | 57.9 | 41.6 |
| | **Gain** | - | +34.5 | - | - | **+26.7** | - |
| | CoT | - | 56.9 | - | - | 40.5 | - |
| Qwen2.5-72B | KARPA | 73.2 | 86.0 | 64.5 | 61.1 | 63.6 | 52.7 |
| | **Gain** | - | +29.1 | - | - | +23.1 | - |

Table 17: Performance comparison of CoT and KARPA methods across different base models (Qwen2.5-7B, Qwen2.5-14B, Qwen2.5-72B) on WebQSP and CWQ datasets.

The results in Table 17 highlight KARPA's ability to operate effectively on unseen KGs by focusing on reasoning and planning rather than leveraging the LLM's pre-existing knowledge. The results also show that KARPA maintained strong performance, even as the LLM's stored knowledge was significantly reduced. This means that even if the LLM does not have ample prior knowledge about a specific domain, KARPA can still leverage the LLM's reasoning and planning capabilities to construct reasoning chains to find the correct answers within the KG.

To demonstrate the effectiveness of KARPA in noisy KGs and specialized domains, we conduct an experiment introducing noise into the KG. For WebQSP and CWQ samples with reasoning paths longer than one, we randomly shuffle the neighboring relations of topic entity and then compared the performance of KARPA and ToG using GPT-4o-mini.

| Knowledge Graphs | Method | WebQSP | | | CWQ | | |
|---|---|---|---|---|---|---|---|
| | | Accuracy | Hit@1 | F1 | Accuracy | Hit@1 | F1 |
| Original KGs | ToG | 54.2 | 72.8 | 50.3 | 47.6 | 52.5 | 39.1 |
| Shuffled KGs | ToG | 32.7 | 48.2 | 30.1 | 23.3 | 26.7 | 20.9 |
| **Variation** | ToG | -21.5 | -24.6 | -20.2 | -24.3 | -25.8 | -18.2 |
| Original KGs | KARPA | 72.3 | 86.4 | 67.2 | 64.6 | 67.7 | 55.1 |
| Shuffled KGs | KARPA | 70.7 | 84.1 | 64.5 | 56.0 | 61.3 | 51.5 |
| **Variation** | KARPA | -1.6 | -2.3 | -2.7 | -8.6 | -6.4 | -3.6 |

Table 18: Comparison of performance between original and shuffled KGs for ToG and KARPA methods on WebQSP and CWQ datasets.

The results in Table 18 show that KARPA experiences a slight drop in performance, demonstrating its resilience to noisy relations. ToG shows a more significant decline, highlighting the limitations of traditional KGQA methods in noisy environments.

To further illustrate KARPA's advantage, we conduct additional experiments comparing training-based method (RoG with fine-tuned LLaMa2-7B) with KARPA using the Qwen-series LLMs (untrained). Both approaches used Qwen LLMs for final answer reasoning.

The results in Table 19 show that while RoG's performance plateaued as the LLM's size and ability increased, KARPA's performance consistently improved, demonstrating its scalability and adaptability. This indicates that KARPA's reliance on pretrained LLMs allows it to benefit from future improvements in LLM reasoning and planning capabilities without requiring retraining.

| Base-Model | Method | WebQSP | | | CWQ | | |
|---|---|---|---|---|---|---|---|
| | | Accuracy | Hit@1 | F1 | Accuracy | Hit@1 | F1 |
| LLaMa2-7B + Qwen2.5-7B | RoG | 54.5 | 73.8 | 57.2 | 38.6 | 43.5 | 35.8 |
| Qwen2.5-7B | KARPA | 65.6 | 79.2 | 58.6 | 47.6 | 52.7 | 38.8 |
| LLaMa2-7B + Qwen2.5-14B | RoG | 58.7 | 77.2 | 60.9 | 43.9 | 48.0 | 42.5 |
| Qwen2.5-14B | KARPA | 72.6 | 84.1 | 65.0 | 51.5 | 57.9 | 41.6 |
| LLaMa2-7B + Qwen2.5-72B | RoG | 57.9 | 76.0 | 59.2 | 45.0 | 50.7 | 43.8 |
| Qwen2.5-72B | KARPA | **73.2** | **86.0** | **64.5** | **61.1** | **63.6** | **52.7** |

Table 19: Comparison between training-based method (RoG) and KARPA using different base-model.

# E  FURTHER DISCUSSION

## E.1  LLM CALL FREQUENCY

KARPA utilizes LLMs in three steps: initial planning, re-planning, and reasoning. However, the re-planning step often generates multiple candidate paths, especially for complex questions or when there are multiple topic entities. Each of these candidate paths is matched to paths within the KG using semantic similarity to retrieve the most relevant reasoning paths. In the reasoning step, the top-K retrieved paths of each candidate paths are provided to the LLM in batches to generate the final answers. As the complexity of the query increases (e.g., in the CWQ dataset), the number of topic entities and candidate paths also increases. Consequently, the number of LLM calls during the reasoning step rises.

In Table 3, we observe that the CWQ dataset requires more LLM calls compared to WebQSP due to its more complex query logic. However, compared to methods that relies on direct interation between LLMs and KGs such as ToG, where LLM call frequency increases significantly with question complexity, KARPA demonstrates much more stable scaling. For instance, in Table 3, ToG requires an average of 3.1 additional calls for the CWQ dataset, while KARPA requires only 0.5 additional calls when using GPT-4o and GPT-4.

## E.2  EFFECTIVENESS BEYOND KGQA TASKS

While KARPA is currently designed to address challenges in KGQA tasks, following the settings of prior works such as RoG and ToG, its methodology is generalizable to other knowledge-intensive tasks.

KARPA's core idea lies in letting LLMs generate complete reasoning chains instead of disrupting reasoning continuity with step-by-step searching. This approach mimics human reasoning processes and enhances reasoning efficiency. For example, in knowledge-intensive task such as the retrieval of academic papers, KARPA could generate reasoning chains like "research field → target journal/conference → specific keywords", and then retrieve the corresponding paper using semantic similarity. When extracting information from books, the reasoning chain like "book title → relevant chapter → relevant paragraphs" could streamline the information retrieval. This reasoning-chain generation aligns with human thought processes, making it both intuitive and adaptable to diverse knowledge-intensive tasks.

## E.3  INCORPORATING USER FEEDBACK MECHANISMS

KARPA's architecture is inherently well-suited to incorporating user feedback mechanisms due to its design of generating complete reasoning paths. Here is a potential extension:

- Initial Path Generation: KARPA generates an initial reasoning path based on the user query.

- Ambiguity Threshold: Using our semantic similarity-based retrieval method, we match the LLM-generated path with paths within the KG. If the similarity score reaches a certain ambiguity threshold, the query is considered clear; if the similarity score falls below that threshold, we identify the query as potentially ambiguous.

- User Feedback: If the similarity score reaches the threshold, we can provide the user with the retrieved answers. If the score falls below the threshold, we could present the extracted reasoning paths to the user for review and request further clarification or refinement of the query.

- Refinement and Re-Retrieval: Based on user feedback, KARPA could adjust the reasoning path and re-run the retrieval process to generate more accurate results.

Through the steps outlined above, KARPA can establish a comprehensive user feedback mechanism, which enhances the precision of queries based on ongoing user feedback.

## F  DETAILED RELATED WORK

### F.1  PROMPT-BASED QUESTION ANSWERING USING INTERNAL KNOWLEDGE

In the field of large language models (LLMs), researchers explore how to combine internal knowledge with external information to enhance reasoning abilities. Existing models utilize a vast internal knowledge base and achieve significant progress in reasoning tasks. To further optimize these capabilities, researchers propose various prompt-based methods, such as Chain of Thought (CoT) (Li et al., 2023c) prompting. This method breaks down complex tasks into manageable steps, promoting structured reasoning and excelling in mathematical and logical reasoning. Building on CoT, researchers also develop variants like Auto-CoT (Zhang et al., 2022), Zero-Shot-CoT (Kojima et al., 2022), Complex-CoT (Fu et al., 2022), and new frameworks such as Tree of Thoughts (ToT) (Yao et al., 2024), which further expand the application range of LLMs.

Additionally, with regard to the "decoding" problem of the reasoning process, Self-consistency CoT (Wang et al., 2022) serves as a representative method. It generates multiple reasoning paths through manually designed prompts and employs a "majority voting" mechanism to identify the "most consistent" path, thereby enhancing CoT performance. CoT verification (Weng et al., 2022) is another important research direction that allows models to self-verify the correctness of their answers through multiple rounds of reasoning. Self-Verification samples multiple candidate reasoning paths and ranks them based on whether the conditions satisfy the conclusions. Recently, OpenAI launches the o1 series models, marking a significant advancement in LLM reasoning abilities, allowing models to develop extensive internal chains of thought and further tap into their reasoning potential.

### F.2  EMBEDDING MODELS AND EMBEDDING-BASED METHODS.

**Embedding models.** Embedding models have revolutionized how we represent and understand text by converting words and sentences into dense vector representations (Mikolov et al., 2013). These embedding models capture the semantic meaning of the text, enabling models to effectively measure the similarity and relationships between different texts. In recent years, significant progress has been made in the field of text embeddings, largely due to the emergence of pre-trained language models (Vaswani et al., 2017). Models like BERT (Devlin et al., 2018) and its variants have become fundamental tools for efficiently encoding the underlying semantics of data. Key advancements in contrastive learning (Xiong et al., 2020), particularly improvements in negative sampling and knowledge distillation applications (Hofstätter et al., 2021), also contribute significantly to the progress in this field. As a result, there is a growing trend to develop universal embedding models that can uniformly support a variety of applications, ranging from information retrieval to natural language processing tasks. Prominent emerging embedding models include Contriever (Izacard et al., 2021), LLM-Embedder (Zhang et al., 2023a) and Open Text Embedding (Neelakantan et al., 2022). These models significantly advance the application of text embeddings across various general tasks.

### F.3 KNOWLEDGE GRAPHS AND RETRIEVAL-AUGMENTED METHODS.

Knowledge graphs and retrieval-augmented generation (RAG) (Lewis et al., 2020) play a crucial role in enhancing various downstream tasks, such as question answering, text generation, and information retrieval. Early research Sun et al. (2018) uses random walk algorithms to retrieve information from knowledge graphs. Subsequent studies Li et al. (2023a); Yu et al. (2021) employ BM25 and DPR algorithms for knowledge graph-based information retrieval, further improving the performance of LLMs. UniKGQA Jiang et al. (2022) integrates the retrieval process with LLMs to achieve state-of-the-art performance in knowledge graph question-answering tasks. GraphRAG Edge et al. (2024) designs a powerful process that extracts structured data from unstructured text using LLMs. These studies collectively demonstrate that information retrieved from knowledge graphs significantly enhances the reasoning capabilities of LLMs. KELP (Liu et al., 2024) utilizes an embedding model to filter reasoning paths from the KG. However, it does not leverage the reasoning capabilities of LLMs and is limited to reasoning paths within a 2-hop range, restricting its applicability to more complex queries. KnowledgeNavigator (Guo et al., 2024a) employs an iterative process where the LLM retrieves and filters relevant knowledge directly from the KG, while Paths-over-Graph (PoG) (Tan et al., 2024) enhances the reliability of LLM-based reasoning by leveraging KG pruning and subgraph reasoning. However, similar to ToG, both methods remain fully dependent on repeated interactions between the LLM and KG, which can result in high computational overhead. LightRAG (Guo et al., 2024b) capitalizes on graph structures by combining LLM-based text indexing with a two-layer retrieval mechanism, improving its capability to integrate information across diverse sources.

## G DATASETS

We adopt two widely-used multi-hop KGQA datasets in our work. Table 20 below gives detailed statistical information for both datasets.

- **WebQuestionsSP (WebQSP)** (Yih et al., 2016) is a knowledge base Q&A dataset containing 4737 questions requiring up to 2-hop reasoning on the KG Freebase (Bollacker et al., 2008), designed to improve the performance of Q&A systems through semantic parsing.
- **Complex WebQuestion (CWQ)** (Talmor, 2018) is extended based on the WebQSP dataset that require up to 4-hop reasoning on the KG Freebase (Bollacker et al., 2008) to solve more complex Q&A tasks.

| Statistics | WebQSP | CWQ |
|---|---|---|
| **Dataset Split** | | |
| Train | 2,826 | 27,639 |
| Test | 1,628 | 3,531 |
| **Question Hop Distribution** | | |
| 1 hop | 65.49% | 40.91% |
| 2 hop | 34.51% | 38.34% |
| $\geq$ 3 hop | 0.00% | 20.75% |
| **Answer Counts Distribution** | | |
| Ans = 1 | 51.2% | 70.6% |
| $2 \leq$ Ans $\leq 4$ | 27.4% | 19.4% |
| $5 \leq$ Ans $\leq 9$ | 8.3% | 6.0% |
| Ans $\geq 10$ | 12.1% | 4.0% |

Table 20: Comprehensive Statistics of Datasets.

## H BASELINES

We consider the following baseline methods for performance comparison:

- **IO Prompt**: Directly query large language models (LLMs) for answers without relying on external sources of information or additional reasoning processes.

- **CoT Prompt**: Utilizing Chain-of-Thought prompting with LLMs to facilitate reasoning involves guiding the LLM through a step-by-step process, where each step reflects the logical sequence of human reasoning.

- **Traning-Based Methods**:

  **KD-CoT** (Wang et al., 2023) interacts with external knowledge to verify and amend the reasoning paths within the Chain-of-Thought (CoT), effectively overcoming issues of hallucinations and error propagation. It structures the CoT reasoning process of LLMs into a formatted multi-round QA approach. In each round, LLMs interact with a QA system that retrieves external knowledge, constructing more reliable reasoning paths based on the precise answers retrieved, thereby enhancing the accuracy and credibility of reasoning.

  **UniKGQA** (Jiang et al., 2022) unifies retrieval and reasoning in both model architecture and parameter learning by designing a shared pre-training task based on question-relation matching and applying fine-tuning strategies to optimize the retrieval and reasoning processes. It includes two main modules: a semantic matching module based on a pre-trained language model (PLM) for question-relation semantic matching, and a matching information propagation module that spreads matching information along directed edges in the knowledge graph (KG).

  **DECAF** (Yu et al., 2022) arrives at the final answer by co-generating logical forms and direct answers and combining the best of both. Unlike approaches that rely on entity linking tools, DECAF simplifies the process of information retrieval by linearizing the knowledge base into text documents and locating relevant subgraphs using text-based retrieval methods.

  **RoG** (Luo et al., 2023) is an approach that combines LLMs with KG to achieve reliable and interpretable reasoning. The method first generates knowledge graph-based relational paths that serve as faithful reasoning plans, and then utilizes these plans to retrieve valid reasoning paths from the knowledge graph for accurate reasoning in LLMs. RoG enhances the reasoning capabilities of LLMs by training to distill knowledge from knowledge graphs and allows them to be seamlessly integrated with arbitrary LLMs for reasoning.

- **Training-Free Methods**:

  **ToG** (Sun et al., 2023) proposes a new LLM-KG integration paradigm "LLM $\otimes$ KG" that treats a LLM as an agent that performs a beam search over the knowledge graph iteratively to discover the most promising reasoning paths and return the most possible reasoning results. ToG leverages the reasoning power of LLMs and expert feedback to ensure traceability and correctability of knowledge. The framework is flexible and plug-and-play for different LLMs, knowledge graphs, and cueing strategies with no additional training cost.

# I PROMPTS

Our proposed KARPA framework consists of the following three main steps: (1) Pre-Planning; (2) Retrieving; (3) Reasoning. Among them, steps (1) and (3) use the Large Language Model (LLM), and Appendix I provides the related Prompts.

## I.1 PRE-PLANNING

### I.1.1 INITIAL-PLANNING PROMPT

In the pre-planning stage, initial planning involves using an LLM to preliminarily generate several relation paths of different lengths. The prompt used for this process is given in Content I.1.1.

---

**✚ Initial-Planning Prompt**

In the process of answer retrieval using a knowledge graph, please think step-by-step and generate reasoning paths of lengths 1, 2, and 3 from a given question and the provided head entity (or entities) that could potentially lead to answer entities. If a reasoning path of the specified length does not exist, please explain the reason.

**Q**:
  Name the president of the country whose main spoken language was Brahui in 1980?
  Topic Entity: Brahui Language
**A**:
  Length 1 reasoning path: The answer entity cannot be reached within a single step, so the length 1 reasoning path is None: {}.
  Length 2 reasoning path: The answer entity may be reached by first finding the corresponding country through the relation "language.human_language.main_country", and then finding the president of the country through the relation "government.government_position_held.office_holder". So the length 2 reasoning path is: {language.human_language.main_country, government.government_position_held.office_holder}.
  Length 3 reasoning path: The answer entity does not require 3 steps to reach, so the length 3 reasoning path is None: {}.

**Q**:
  Who is Tom's wife?
  Topic Entity: Tom
**A**:
  Length 1 reasoning path: The answer entity can be reached within a single step by finding Tom's spouse through the relation "people.person.spouse_s". Therefore, the length 1 reasoning path is: {people.person.spouse_s}.
  Length 2 reasoning path: The answer entity of the question may be reached if we first find the children through first relation "people.person.children", and then find the parent through second relation "people.person.parent". Therefore, the length 2 reasoning path is: {people.person.children, people.person.parent}.
  Length 3 reasoning path: The answer entity of the question does not require 3 steps to reach, so the length 3 reasoning path is None: {}.

**Q**:
  {*A Question.*}
  Topic Entity: {*An Entity*}
**A**:

### I.1.2 RE-PLANNING PROMPT

In the re-planning of pre-planning, the LLM is used to re-plan relation paths based on the retrieved relations (specifically the top-$K$ relations), which are then used as retrieval information in the retrieving step. The prompt used is shown in Content I.1.2.

---

**📖 Re-Planning Prompt**

Given a set of relations and a question, please select relevant relations from the provided relation set to form reasoning paths of length 1, 2, and 3 that could lead from the provided topic entity (or entities) to potential answer entities in a knowledge graph. Ensure that the reasoning paths you create logically connect the topic entity and potential answer entities. Only consider length 3 paths if shorter paths are insufficient to reach the answer. If a reasoning path of the specific length cannot be formed, please explain why.

**Q**:
  Name the president of the country whose main spoken language was Brahui in 1980?
  Topic Entity: Brahui Language
  Relations:
    language.human_language.language_family; language.human_language.main_country;
    base.rosetta.languoid.parent; language.human_language.writing_system;
    language.human_language.countries_spoken_in; kg.object_profile.prominent_type;

base.ontologies.ontology_instance.equivalent_instances;
government.government_position_held.office_holder; language.human_language.region;

**A**:

Length 1 reasoning path: The provided relations cannot reach the answer entity in one step, so the length 1 reasoning path is None: .

Length 2 reasoning path: The answer entity may be reached by first finding the corresponding country through the provided relation "language.human_language.main_country", and then finding the president of the country through the relation "government.government_position_held.office_holder". So the length 2 reasoning path is: language.human_language.main_country, government.government_position_held.office_holder.

Length 3 reasoning path: The answer entity does not require 3 steps to reach, so the length 3 reasoning path is None: .

**Q**:

Who is Tom's wife?
Topic Entity: Tom
Relations:

people.person.profession; people.marriage.spouse; people.person.nationality;
award.award_nomination.award_nominee; people.person.parents;
award.award_nominee.award_nominations; people.person.children;

**A**:

Length 1 reasoning path: Tom's wife in knowledge graph could be reached within a single step by finding Tom's spouse through the provided relation "people.person.spouse_s". Therefore, the length 1 reasoning path is: people.person.spouse_s.

Length 2 reasoning path: Tom's wife may be reached if we first find the children through the relation "people.person.children", and then find the parent through second relation "people.person.parent". Therefore, the length 2 reasoning path is: people.person.children, people.person.parent.

Length 3 reasoning path: The answer entity of the question does not require 3 steps to reach, so the length 3 reasoning path is None: .

**Q**:

{*A Question.*}
Topic Entity: {*An Entity.*}
Relations: {*A list of Relations.*}

**A**:

## I.2 REASONING

In the reasoning step, the top-$K$ relation paths retrieved in the retrieving step, along with their connected topic entity, answer entities, the corresponding question, and all related information are input into the LLM. The prompt used is provided in content I.2 below.

---

**⚓ Reasoning Prompt**

Given a question and the associated retrieved knowledge graph reasoning paths (topic entity, relation path, tail entity/entities), please think step-by-step and determine whether the tail entity/entities of each provided reasoning paths are the right answer to the question. If any of the tail entities are the correct answer, return those tail entities in the format of answer_1, answer_2, .... If the tail entity of current reasoning path is not the answer of question, explain why. All reasoning paths should be analysed.

**Q**:

Rift Valley Province is located in a nation that uses which form of currency?
Reasoning Paths:

(Rift Valley Province, location.administrative_division.country, Kenya)
(Rift Valley Province, location.location.geolocation, UnName_Entity)
(Rift Valley Province, location.administrative_division.country
→ location.country.currency_used, Kenyan shilling)

**A**:

Let's analyze the reasoning paths step-by-step to determine the correct answer to the question.

1.(Rift Valley Province, location.administrative_division.country, Kenya): Rift Valley Province is located in Kenya. However, it does not provide information about the currency used in Kenya, so Kenya is not the direct answer to the question.

2.(Rift Valley Province, location.location.geolocation, UnName_Entity): The UnName_Entity is not the answer of question.

3.(Rift Valley Province, location.administrative_division.country→ location.country.currency_used, Kenyan shilling): Kenyan shilling is the form of currency used in the nation where Rift Valley Province is located, so 'Kenyan shilling' is the right answer to the question.

The correct answer to the question is the Kenyan shilling, as identified in the fourth reasoning path. Therefore, the correct tail entity is: Kenyan shilling.

**Q**:
 {*A Question.*}
 Reasoning Paths: {*A list of Reasoning Paths.*}
**A**:

## J    CASE STUDIES

In this section, we present a detailed case study to illustrate the effectiveness of KARPA in handling complex knowledge graph question answering (KGQA) tasks. KARPA leverages LLMs in both the pre-planning and reasoning steps. For the question "What did James K. Polk do before he was president?", KARPA uses the LLM to generate initial reasoning paths and then further refines the answer by reasoning over the identified relation paths and corresponding entities. The following case study elaborates on the workflow of KARPA in this example, showcasing its ability to utilize external knowledge and LLM planning capabilities to accurately answer the question.

**Initial-Planning**

**Q:**
what did james k polk do before he was president?
Topic Entity: James K. Polk

**Response of GPT4:**

Length 1 reasoning path: The answer entity cannot be reached with-in a single step, so the length 1 reasoning path is None: {}.

Length 2 reasoning path: The answer entity may be reached by first finding the political positions held by James K. Polk through the relation "government.politician.position_held", and then filtering for the position held before he became president. So the length 2 reasoning path is:
{government.politician.position_held,
filter: position before president}.

Length 3 reasoning path: The answer entity may be reached by first finding the political positions held by James K. Polk through the relation "government.politician.position_held", then finding the specific position such as "Speaker of the House" or "Governor" through the relation "government.government_office_or_title.office_holder", and finally filtering for the position held before he became president.
So the length 3 reasoning path is:
{government.politician.position_held,
government.government_office_or_title.office_holder,
 filter: position before president}.

Figure 4: A Case for Relational Path Generation in the Initial-Planning Phase.

In the pre-planning step, KARPA first utilizes the LLM to generate initial relational paths based on the provided question, as shown in Figure 4. Given the question "What did James K. Polk do before he was president?", the LLM generates paths of varying lengths. Initially, the LLM considers whether the answer entities can be reached within a single relational step. Since the LLM considers the answer entities for this question cannot be reached in one step, the LLM outputs an empty reasoning path of length 1.

When considering a relational path with two associated relations, the LLM infers that the answer entity can be found by first identifying the political positions held by James K. Polk through the relation "government.politician.position_held," and then filtering for the position he held before becoming president using "filter: position before president." Thus, the LLM determines that the answer entities can be reached via the path {government.politician.position_held, filter: position before president}. Additionally, the LLM considers that the answer entities might be accessible through a path involving three relations. This step-by-step reasoning process allows the LLM to initially plan multiple reasoning chains for subsequent relation retrieval.

---

**🖥️ Re-Planning**

**Q:**

What did james k polk do before he was president?
Topic Entity: James K. Polk
Relations:
government.politician.government_positions_held;
government.government_position_held.office_holder;
government.government_position_held.appointed_by;
government.government_position_held.basic_title;
government.government_position_held.governmental_body;
government.us_president.vice_president;
people.appointee.position;
government.us_vice_president.to_president;
government.government_office_or_title.office_holders;
government.government_position_held.office_position_or_title;
government.government_office_or_title.jurisdiction;
government.government_office_or_title.governmental_body_if_an
y; base.inaugurations.inauguration.president;
military.military_person.participated_in_conflicts;
government.politician.election_campaigns

**Response of GPT4:**

Length 1 reasoning path: The provided relations cannot reach the answer entity in one step, so the length 1 reasoning path is None: {}.

Length 2 reasoning path: The answer entity may be reached by first finding the government positions held by James K. Polk through the relation "government.politician.government_positions_held", and then identifying the specific position held before becoming president through the relation "government.government_position_held.office_position_or_title". Therefore, the length 2 reasoning path is: {government.politician.government_positions_held, government.government_position_held.office_position_or_title}.

Length 3 reasoning path: The answer entity does not require 3 steps to reach, so the length 3 reasoning path is None: {}.

Figure 5: A Case of Relation Path Assembling in the Re-Planning Phase.

In the third phase of the pre-planning step, KARPA employs the LLM to re-plan the relational paths based on the set of extracted relations. For the question "What did James K. Polk do before he was

president?", the LLM is provided with a set of relations, as illustrated in Figure 5. The LLM is tasked with selecting relevant relations from the list and assembling them into complete reasoning chains that potentially connect the topic entity to the answer entities.

In this case, the LLM determines that the answer entities cannot be reached using a single relation from the provided list, and therefore outputs an empty relation path for length 1. When constructing a relation path of length 2, the LLM identifies that "government.politician.government positions held" and "government.government position held.office position or title" form a complete reasoning chain, enabling the extraction of the correct answer entities for the given question. As a result, the LLM outputs the length 2 relation path as {government.politician.government_positions_held, government.government_position_held.office_position_or_title}. Since the LLM considers that the answer can be retrieved using this two-step reasoning chain, it determines that a three-step reasoning chain is unnecessary and outputs None for the length 3 relation paths.

In the reasoning step of KARPA, several candidate relational paths are provided for the LLM to determine the final answer. Given these candidate paths and their corresponding entities, the LLM analyzes each path step-by-step, enabling more thoughtful and accurate reasoning. An example of KARPA's reasoning process is illustrated in Figure 6.

> ### 🧠 Reasoning
>
> **Q:**
> what did james k polk do before he was president?
> Reasoning Paths:
> ('James K. Polk', 'government.politician.government_positions_held ->
> government.government_position_held.basic_title', 'President, Speaker')
> ('James K. Polk', 'government.politician.government_positions_held ->
> government.government_office_or_title.office_holders', 'Speaker of the United States House of
> Representatives')
> ('James K. Polk', 'government.politician.government_positions_held ->
> government.government_position_held.governmental_body', 'United States House of Representatives')
> ('James K. Polk', 'government.government_position_held.office_holder ->
> government.government_position_held.office_position_or_title', 'United States Representative')
> ('James K. Polk', 'government.politician.government_positions_held ->
> government.government_position_held.legislative_sessions', '25th United States Congress, 24th United
> States Congress')
>
> **Response of GPT4:**
>
> Let's analyze each reasoning path to deduce if it answers the question regarding what James K. Polk
> did before he became president.
>
> 1. ('James K. Polk', 'government.politician.government_positions_held ->
> government.government_position_held.basic_title', 'President, Speaker'):
>     - This path mentions "President, Speaker" indicating he held the position of Speaker possibly before
> becoming President. "Speaker" could potentially be a valid answer but it lacks the detail of which
> Speaker position.
>
> 2. ('James K. Polk', 'government.politician.government_positions_held ->
> government.government_office_or_title.office_hold
> -ers', 'Speaker of the United States House of Representatives'):
>     - This path clearly states that before becoming President, he was the "Speaker of the United States
> House of Representatives." This directly answers the question about what he did before becoming
> President.
>
> 3. ('James K. Polk', 'government.politician.government_positions_held ->
> government.government_position_held.government
> -tal_body', 'United States House of Representatives'):
>     - This indicates he was a member of the United States House of Representatives, which is relevant
> as one major position held prior to being President but less detailed compared to the Speaker role.
>
> 4. ('James K. Polk', 'government.government_position_held.office_holder -
> >government.government_position_held.office_po-
> sition_or_title', 'United States Representative'):
>     - This details his role as a United States Representative, a position held in the legislature before
> becoming President.
>
> 5. ('James K. Polk', 'government.politician.government_positions_held ->
> government.government_position_held.legislative
> _sessions', '25th United States Congress, 24th United States Congress'):
>     - This confirms his participation in the 24th and 25th United States Congresses, indicating his
> legislative roles.
>
> From these reasoning paths, the valid answers to what James K. Polk did before he was president are:
> {Speaker of the United States House of Representatives, United States Representative}

Figure 6: A Case of Inputting Top-$K$ Relation Paths into LLMs During Reasoning Phase.

