# OpenReview forum: "KARPA: A Training-free Method of Adapting Knowledge Graph as References for Large Language Model's Reasoning Path Aggregation"
_ICLR.cc/2025/Conference — ICLR 2025 Conference Withdrawn Submission_

### Official Review · Reviewer_oYxY · 2024-10-23

**Soundness:** 2
**Presentation:** 2
**Contribution:** 2
**Rating:** 5
**Confidence:** 4

**Summary:**

The paper introduces a novel framework called KARPA designed to enhance the reasoning capabilities of LLMs in Knowledge Graph Question Answering (KGQA) tasks. The authors identify limitations in existing methods that either rely on step-by-step traversal of KGs, which restricts the global planning abilities of LLMs, or require fine-tuning on specific KGs, making them less adaptable.

KARPA addresses these challenges through a three-step process:
(1) Pre-planning: The LLM generates initial relation paths based on the question and relevant KG relations, leveraging its inherent global reasoning and planning capabilities.
(2) Retrieving: A semantic embedding model is used to find candidate paths in the KG that are semantically similar to the LLM-generated paths, avoiding local optima and reducing interactions with the KG.
(3) Reasoning: The candidate paths and corresponding entities are provided back to the LLM for comprehensive reasoning to produce the final answer.

The framework operates in a training-free manner, making it adaptable to various LLM architectures without additional fine-tuning or pre-training. Experimental results demonstrate that KARPA achieves state-of-the-art performance on multiple KGQA benchmark datasets, delivering both high efficiency and accuracy.

**Strengths:**

- By eliminating the need for fine-tuning or pre-training on specific KGs, KARPA offers a flexible and adaptable solution compatible with various LLMs.
- KARPA reduces the number of interactions between the LLM and the KG, enhancing efficiency without compromising accuracy. Moreover, the semantic embedding-based retrieval mitigates the risk of the LLM getting trapped in locally optimal solutions, leading to more effective exploration of KGs.
- KARPA fully leverages the LLM's global reasoning and planning abilities, enabling it to generate comprehensive relation paths beyond adjacent relations.

**Weaknesses:**

- The experimental evaluation is focused on KGQA tasks; the framework's effectiveness on other knowledge-intensive tasks remains unaddressed.
- KARPA assumes access to comprehensive and accurate KGs, but the performance may degrade with incomplete or noisy KGs. Further, the effectiveness of KARPA may vary across different domains, especially if the KG lacks sufficient coverage of the required knowledge.
- The retrieval of candidate paths depends heavily on the quality of the semantic similarity measures used in the embedding model, which may affect the overall performance if the embeddings are not optimal.

**Questions:**

- How does KARPA perform when the underlying KG is incomplete or contains inaccuracies? Have you tested the robustness of the framework in such scenarios?
- Have you evaluated the computational efficiency and scalability of KARPA on larger KGs or with more complex queries?
- Can KARPA be extended or adapted to other knowledge-intensive tasks beyond KGQA? If so, how would it perform?
- How does KARPA's performance compare with state-of-the-art models that utilize fine-tuning or pre-training on specific KGs?
- Do you consider ncorporating user feedback mechanisms to handle ambiguous queries or to refine the reasoning paths generated by the LLM. If so, how to make it.

Missing References
- Knowledge Graph-Enhanced Large Language Models via Path Selection (ACL 2024, Findings)
- KnowledgeNavigator: Leveraging Large Language Models for Enhanced Reasoning over Knowledge Graph (Complex & Intelligent Systems, 2024)
- Paths-over-Graph: Knowledge Graph Enpowered Large Language Model Reasoning (2024)
- LightRAG: Simple and Fast Retrieval-Augmented Generation (2024)
- ……

---

> ### Author Response · Authors · 2024-11-26
> **Response to Reviewer oYxY (1)**
>
> We sincerely thank you for your thoughtful review and your recognition of the strengths of
> our work. Below, we address each of your concerns in detail and provide additional evidence
> to support our claims.
>
> ### Weaknesses:
>
> > **W1. Effectiveness Beyond KGQA Tasks**
>
> Thank you for pointing this out. While KARPA is currently designed to address challenges
> in KGQA tasks, following the settings of prior works such as RoG and ToG, its methodology
> is generalizable to other knowledge-intensive tasks.
>
> KARPA’s **core idea** lies in letting LLMs generate complete reasoning chains instead
> of disrupting reasoning continuity with step-by-step searching. This approach **mimics
> human reasoning processes** and enhances reasoning efficiency. For example, in knowledge-intensive
> task such as the retrieval of academic papers, KARPA could generate reasoning chains
> like “research field → target journal/conference → specific keywords”, and then retrieve the
> corresponding paper using semantic similarity. When extracting information from books, the
> reasoning chain like “book title → relevant chapter → relevant paragraphs” could streamline
> the information retrieval.
> This reasoning-chain generation aligns with human thought processes, making it both **intuitive
> and adaptable** to diverse knowledge-intensive tasks. While our current work focuses on KGQA,
> we will discuss the broader applicability of KARPA in the revised paper and outline potential
> applications in the appendix.

---

> > ### Comment · Reviewer_oYxY · 2024-11-29
> >
> > Dear Authors,
> >
> > Thank you very much for the clarification! Some of them have addressed my concerns. But some of them cannot convince me, such as the explanation on "W2. Performance in Specialized Domains", "Q3. Applicability to Other Knowledge-Intensive Tasks" and "W1. Effectiveness Beyond KGQA Tasks".
> >
> > I appreciate the detailed response, especially the newly conducted experiments. Including those in the revised draft will strengthen the paper.
> >
> > Overall, I acknowledge the effort of this work, but still have reservations about viewpoints of paper quality, and decide to maintain my original score.
> >
> > Best Regards,
> >
> > Reviewer oYxY

---

> > > ### Author Response · Authors · 2024-11-29
> > > **Response to Reviewer oYxY (5)**
> > >
> > > Dear Reviewer oYxY,
> > >
> > > Thank you for your thoughtful and detailed response. Your comments have undoubtedly strengthened the clarity and depth of our work. We have incorporated the additional experiments into the revised draft (now included in the Appendix under Additional Experiments as Table 17, Table 18, Table 10, and Table 19). Additionally, we have expanded the Future Discussion section to include considerations on Effectiveness Beyond KGQA Tasks, Other Knowledge-Intensive Tasks, and User Feedback Mechanisms.
> > >
> > > Regarding "W2. Performance in Specialized Domains", we will conduct further experiments to address your concern. For "W1. Effectiveness Beyond KGQA Tasks" and "Q3. Applicability to Other Knowledge-Intensive Tasks", as KARPA is primarily designed to address challenges in KGQA, we acknowledge that our discussion of other tasks might lack depth. Could you kindly provide more details about the aspects or applications you would like us to explore? Your insights would greatly help us refine this discussion further.
> > >
> > > Thank you again for your valuable feedback and the time you have dedicated to reviewing our work. Wishing you a wonderful Thanksgiving!
> > >
> > > Best regards,
> > >
> > > Authors of KARPA

---

> ### Author Response · Authors · 2024-11-26
> **Response to Reviewer oYxY (2)**
>
> > **W2. Performance in Noisy KGs and Specialized Domains**
>
> **Performance in Noisy KGs:**
>
> You have identified an important aspect of KARPA’s design. Most existing KGQA frameworks,
> including RoG and ToG, assume access to structured, reliable knowledge graphs (KGs). However, when
> KGs are noisy or incomplete, these methods often fail because the LLM relies on **faulty or
> misleading relations**, leading to incorrect answers. KARPA addresses this issue through two designs:
> - LLMs generate candidate paths using the **most relevant relations** in the KG,
> avoiding reliance on adjacent or noisy relations.
> - Our heuristic value-based retrieval
> method extracts paths with the highest semantic similarity of **entire reasoning paths**,
> mitigating the effects of noise or incomplete relations.
>
> To validate KARPA’s robustness, we conduct an experiment introducing noise into the KG. For
> WebQSP and CWQ samples with reasoning paths longer than one, we **randomly shuffle**
> the neighboring relations of topic entity and then compared the performance of KARPA and ToG
> using GPT-4o-mini.
>
> | Knowledge Graphs (KGs) | Method | Accuracy (WebQSP) | Hit@1 (WebQSP) | F1 (WebQSP) | Accuracy (CWQ) | Hit@1 (CWQ) | F1 (CWQ) |
> |------------------------|--------|-------------------|----------------|-------------|----------------|-------------|----------|
> | Original KGs           | ToG    | 54.2              | 72.8           | 50.3        | 47.6           | 52.5        | 39.1     |
> | Shuffled KGs           | ToG    | 32.7              | 48.2           | 30.1        | 23.3           | 26.7        | 20.9     |
> | Variation              | ToG    | -21.5             | -24.6          | -20.2       | -24.3          | -25.8       | -18.2    |
> | Original KGs           | KARPA  | 72.3              | 86.4           | 67.2        | 64.6           | 67.7        | 55.1     |
> | Shuffled KGs           | KARPA  | 70.7              | 84.1           | 64.5        | 56.0           | 61.3        | 51.5     |
> | Variation              | KARPA  | -1.6              | -2.3           | -2.7        | -8.6           | -6.4        | -3.6     |
>
> The results show that KARPA experiences a slight drop in performance, demonstrating its **resilience
> to noisy relations**. ToG shows a more significant decline, highlighting the limitations of
> traditional KGQA methods in noisy environments.
>
> **Performance in Specialized Domains:**
>
> KARPA is designed to **minimize** reliance on the LLM's internal knowledge and instead leverage
> its reasoning and planning capabilities to construct logical reasoning chains. To correctly
> answer a question, KARPA must construct a reasoning path that
> aligns with the KG’s structure and leads to the correct answer entity. This means KARPA
> **emphasizes reasoning capability over stored knowledge**. To further validate this, we
> test KARPA with smaller-scale LLMs (Qwen2.5-7B, Qwen2.5-14B) and compared it with Chain of Thought
> (CoT), where LLMs directly answer questions using internal knowledge.
>
> | Base-model  | Method   | Accuracy (WebQSP) | Hit@1 (WebQSP) | F1 (WebQSP) | Accuracy (CWQ) | Hit@1 (CWQ) | F1 (CWQ) |
> |-------------|----------|-------------------|----------------|-------------|----------------|-------------|----------|
> | Qwen2.5-7B  | CoT      | -                 | 41.5           | -           | -              | 28.3        | -        |
> |             | KARPA    | 65.6              | 79.2           | 58.6        | 47.6           | 52.7        | 38.8     |
> |             | **Gain** | -                 | +37.7          | -           | -              | +24.4       | -        |
> | Qwen2.5-14B | CoT      | -                 | 49.6           | -           | -              | 31.2        | -        |
> |             | KARPA    | 72.6              | 84.1           | 65.0        | 51.5           | 57.9        | 41.6     |
> |             | **Gain** | -                 | +34.5          | -           | -              | +26.7       | -        |
>
> The results show that KARPA **maintained strong performance**, even as the LLM’s stored knowledge
> was **significantly reduced**. This means that even if the LLM does not have ample prior knowledge
> about a specific domain, KARPA can still **leverage the LLM's reasoning and planning capabilities**
> to construct reasoning chains to find the correct answers within the KG.
>
> These findings confirm that KARPA can effectively leverage LLMs’
> reasoning abilities even with incomplete or domain-specific KGs. We will include these results
> in our revised manuscript.

---

> ### Author Response · Authors · 2024-11-26
> **Response to Reviewer oYxY (3)**
>
> > **W3. Dependence on Embedding Model Quality**
>
> We appreciate your insightful observation. To examine the impact of different embedding
> models on KARPA’s performance, we conducted experiments using various embeddings.
>
> | Embedding Model                                | Accuracy (WebQSP) | Hit@1 (WebQSP) | F1 (WebQSP) | Accuracy (CWQ) | Hit@1 (CWQ) | F1 (CWQ) |
> |------------------------------------------------|-------------------|----------------|-------------|----------------|-------------|----------|
> | all-MiniLM-L6-v2 (~86MB)                       | 72.3              | 86.4           | 67.2        | 64.6           | 67.7        | 55.1     |
> | all-mpnet-base-v2 (~417MB)                     | 74.5              | 86.1           | 68.6        | 64.1           | 68.3        | 53.7     |
> | paraphrase-multilingual-MiniLM-L12-v2 (~448MB) | 74.1              | 85.3           | 68.3        | 65.3           | 69.5        | 55.4     |
>
> In this table, all-MiniLM-L6-v2 is the embedding model used in our paper, with a size of
> approximately 86MB. all-mpnet-base-v2, a more powerful embedding model, is around 417MB.
> paraphrase-multilingual-MiniLM-L12-v2, which supports embedding between multiple languages,
> has a size of approximately 448MB.
> The results demonstrate that KARPA’s robust design ensures that its overall performance
> remains consistent across different embedding models. This is because the candidate paths generated by KARPA during the pre-planning phase are very distinct. While they are semantically close to the correct reasoning paths, they differ significantly from incorrect reasoning paths. Therefore, a basic embedding model is sufficient to assist KARPA in extracting the correct paths.
> We will add these results into our revised paper.
>
> ### Questions:
>
> > **Q1. Robustness with Incomplete or Inaccurate KGs**
>
> Thank you for raising this important concern. As noted in our response to **Weakness 2**,
> we conduct experiments to evaluate KARPA’s robustness in handling inaccuracies and noise in KGs.
> Specifically, we **randomly shuffle** the relations adjacent to the topic entity in samples where
> the reasoning path lengths are greater than 1 in the WebQSP and CWQ datasets.
> This simulates noise and inaccuracies in the KG. KARPA consistently outperforms
> ToG on the revised datasets, which requires direct LLM-KG interactions.
> This demonstrates that KARPA is more robust in handling noisy KGs due
> to its **pre-planning** and **semantic similarity-based retrieval** methods. These methods mitigate
> the impact of hallucinations and ensure logical reasoning even with noisy KG structures.
>
> > **Q2. Computational Efficiency and Scalability**
>
> KARPA is designed to be efficient and scalable for a variety of KGQA scenarios,
> including larger KGs and more complex queries. In our experiments, the CWQ dataset exhibits
> a **higher level of complexity and logical intricacy** in its queries compared to the WebQSP dataset.
> For instance, while a question in WebQSP is as straightforward as 'What language is
> spoken in Haiti today?', a question in CWQ might be more complex, such as 'What are the most
> common languages spoken in the country where the Aegean cat breed originated?'.
> In the WebQSP dataset, nearly all questions can be answered **within 2-hop reasoning paths**,
> whereas in the CWQ dataset, over 20% of the questions require **at least 3-hop reasoning paths**
> to locate the corresponding answer entities within the KG.
> (More detailed information of these two datasets can be found in the **Datasets** section
> of the appendix in our paper.)
>
> As shown in **Table 3** of our paper, KARPA scales more
> efficiently than direct LLM-KG interaction methods (e.g., ToG). For CWQ, the number of
> LLM calls in ToG increased by **3.1** on average, while KARPA required only **0.5** additional calls
> using both GPT-4o and GPT-4 as backbone LLMs. This demonstrates KARPA’s ability to maintain
> computational efficiency and scalability even with increasing KG size and query complexity.
>
> > **Q3. Applicability to Other Knowledge-Intensive Tasks**
>
> Thank you for this insightful question. While our current work focuses on KGQA following the
> settings of ToG and RoG, KARPA’s principles are broadly applicable to other knowledge-intensive
> tasks. As noted in our response to **Weakness 1**, one of the key innovations of KARPA is its
> approach of enabling LLMs to **generate complete reasoning chains** directly, rather than
> using step-by-step beam search, which often disrupts the reasoning chain's logical consistency.
> This approach has significant implications for tasks where logical consistency across multiple
> steps is critical.
>
> Our results in KGQA demonstrate that KARPA outperforms existing methods by achieving higher
> accuracy and reasoning efficiency. We believe similar benefits would extend to other tasks,
> making KARPA a strong candidate for advancing other knowledge-intensive domains.

---

> ### Author Response · Authors · 2024-11-26
> **Response to Reviewer oYxY (4)**
>
> > **Q4. Comparison with Training-Based Methods**
>
> In **Table 1** of our paper, we compare KARPA with several fine-tuning-based KGQA approaches,
> including RoG, which represents the state-of-the-art training-based method. These approaches
> achieve reasonable results by injecting KG knowledge into LLMs. However, they also face
> limitations, including **susceptibility to hallucinations** and challenges in **adapting to
> unseen KGs** (see **Figure 1(a)** in our paper). To further illustrate KARPA’s advantage, we conduct
> additional experiments comparing RoG (fine-tuned LLaMa2-7B) with KARPA using the
> Qwen-series LLMs (untrained). Both approaches used Qwen LLMs for final answer reasoning.
>
> | Base-model              | Method | Accuracy (WebQSP) | Hit@1 (WebQSP) | F1 (WebQSP) | Accuracy (CWQ) | Hit@1 (CWQ) | F1 (CWQ) |
> |-------------------------|--------|-------------------|----------------|-------------|----------------|-------------|----------|
> | LLaMa2-7B + Qwen2.5-7B  | RoG    | 54.5              | 73.8           | 57.2        | 38.6           | 43.5        | 35.8     |
> | Qwen2.5-7B              | KARPA  | 65.6              | 79.2           | 58.6        | 47.6           | 52.7        | 38.8     |
> | LLaMa2-7B + Qwen2.5-14B | RoG    | 58.7              | 77.2           | 60.9        | 43.9           | 48.0        | 42.5     |
> | Qwen2.5-14B             | KARPA  | 72.6              | 84.1           | 65.0        | 51.5           | 57.9        | 41.6     |
> | LLaMa2-7B + Qwen2.5-72B | RoG    | 57.9              | 76.0           | 59.2        | 45.0           | 50.7        | 43.8     |
> | Qwen2.5-72B             | KARPA  | 73.2              | 86.0           | 64.5        | 61.1           | 63.6        | 52.7     |
>
> The results show that while RoG’s performance **plateaued** as the LLM’s size and ability increased,
> KARPA’s performance **consistently improved**, demonstrating its scalability and adaptability.
> This indicates that KARPA’s reliance on pretrained LLMs allows it to benefit from future
> improvements in LLM reasoning and planning capabilities without requiring retraining.
>
> > **Q5. Incorporating User Feedback Mechanisms**
>
> Thank you for this excellent suggestion. KARPA’s architecture is inherently well-suited
> to incorporating user feedback mechanisms due to its design of generating complete
> reasoning paths. Here is a potential extension:
> 1. **Initial Path Generation:** KARPA generates an initial reasoning path based on the user query.
> 2. **Ambiguity Threshold:** Using our semantic similarity-based retrieval method,
> we match the LLM-generated path with paths within the KG. If the similarity score reaches
> a certain ambiguity threshold, the query is considered clear;
> if the similarity score falls below that threshold, we identify the query as potentially ambiguous.
> 3. **User Feedback:** If the similarity score reaches the threshold, we can provide the user with the
> retrieved answers. If the score falls below the threshold, we could present the extracted
> reasoning paths to the user for review and request further clarification or refinement of the query.
> 4. **Refinement and Re-Retrieval:** Based on user feedback, KARPA could adjust the reasoning
> path and re-run the retrieval process to generate more accurate results.
>
> Through the steps outlined above, we believe that KARPA can establish a comprehensive user
> feedback mechanism, which enhances the precision of queries based on ongoing user feedback.
>
> > **Missing References**
>
> Thank you for bringing these references to our attention. Although KARPA has substantial
> differences from these works, we acknowledge their relevance to the field of LLM-based KGQA. We will
> incorporate these references into the updated version of our paper, clearly distinguishing their
> methodologies and contributions from those of KARPA.
>
> We are grateful for your constructive feedback, which has helped us refine our work and
> expand its scope. Should our responses have adequately met your concerns, we respectfully ask a review of the rating. We eagerly anticipate any additional perspectives you might share and remain grateful for your thorough examination of our work.

---

### Official Review · Reviewer_Uom3 · 2024-10-28

**Soundness:** 2
**Presentation:** 2
**Contribution:** 2
**Rating:** 5
**Confidence:** 4

**Summary:**

The paper addressed the task of KGQA and focused on the LLM-based method. The paper proposed a knowledge graph-assisted reasoning path aggregation approach, which contains three main steps: pre-planning, retrieving and reasoning. The proposed approach employs LLM and heuristic value-based relation path retrieval to extract potential relational paths for the questions. At last, the extracted candidate paths are provided to the LLM for reasoning. The results show the effectiveness of the proposed approach.

**Strengths:**

1. The proposed heuristic value-based relation path retrieval method is interesting to compute the semantic similarity between paths with different lengths.
2. The experimental comparisons with baselines on KGQA show the effectiveness.

**Weaknesses:**

1) The Figure 1 is not very clear. Totally, there are two limitations for different previous methods: 1) local search strategies and 2) struggle with unseen KGs. I am not sure how the proposed method could resolve the aforementioned limitations. In specific, how about the performance of unseen KGs? Why the simple embedding-based semantic similarity, such as beam search deduce suboptimal answers?

2) The advantage of the proposed heuristic value-based relation path retrieval method is to compute the semantic similarity between paths with different lengths. Why do we need to extend relation paths with different lengths? The authors should exploit some experiments to prove it.

3) I wonder how about the performance when the given KG is not in the scope of LLM training sets. The used WebQSP and CWQ datasets, in my opinion, have been seen when pre-training LLMs.

**Questions:**

See the weakness.

---

> ### Author Response · Authors · 2024-11-26
> **Response to Reviewer Uom3 (1)**
>
> We sincerely thank you for your thoughtful review and recognition of our work's strengths. Below, we provide detailed responses to your comments and address the questions you raised.
> ### Weaknesses:
> > **W1. Addressing Two Limitations for Previous Methods**
>
> Thank you for raising these important questions. Addressing these two limitations was indeed the motivation behind KARPA. The detailed process of KARPA can be found in the **Approach** section of our paper, and we provide **further clarifications** here:
>
> **1. Performance on Unseen KGs:**
>    - **Limitation in Training-Based Methods:**
> Training-based approaches that fine-tune LLMs on specific KGs often fail to generalize to unseen KGs. These methods generate incorrect reasoning paths due to their reliance on prior knowledge from the training data.
>    - **How KARPA Overcomes This Limitation:**  \
> In contrast, KARPA is entirely **training-free**, and all KGs it interacts with are treated as unseen KGs. Instead of relying on prior knowledge of the KG, KARPA combines the capabilities of an embedding model and the LLM to ensure robust reasoning across any KG. Specifically:
>      - **Pre-Planning Stage:** In the pre-planning stage, KARPA uses the embedding model to extract the most relevant relations from the KG, guiding the LLM in **constructing candidate relation paths using relations within the KG**. By doing so, we avoid the risk of the LLM generating paths that deviate from the KG’s structure.
>      - **Retrieval Stage:** In the retrieval stage, even if the candidate paths generated by the LLM are not perfectly aligned with the actual KG structure, our semantic similarity-based retrieval methods ensure that the true reasoning paths from the KG are identified. These paths are both **semantically closest** to the LLM-generated paths and **structurally valid** in the KG.
>      - **Collaborative Process:** This process between the embedding model and LLM ensures errors in the reasoning chain are corrected, allowing KARPA to perform reliably on unseen KGs.
>
> **2. Suboptimal Answers from Local Search Strategies:**
>    - **Limitation in Local Search Strategies:**
> As shown in **Figure 1(b)** of our paper, in methods where the LLM directly interacts with the KG **step-by-step**, the LLM selects the top-K relations at each step based on their likelihood of pointing to the answer entity. If the LLM fails to select the correct relation in any step, it risks completely missing the correct answer entity. For instance, in **Figure 1(b)**, if the LLM misses the relation "CEO of" in the first step, it will fail to reach the correct answer entity, "Priscilla Chan". KARPA with simple embedding similarity-based beam search suffer from similar issues. We treat semantic similarity as a **cost metric** and extract paths using beam search. However, when facing the KG in **Figure 2**, if a critical relation like "CEO of" is associated with a higher cost, beam search may fail to include it, leading to suboptimal answers.
>    - **How KARPA Addresses This Limitation:**
> KARPA addresses these issues through **pathfinding-based (Dijkstra) retrieval methods** and **heuristic value-based retrieval method**. Unlike step-by-step local search strategies, KARPA uses an embedding model to compute the semantic similarity between the LLM-generated paths and paths within the KG. This approach allows KARPA to evaluate the **entire reasoning path** holistically, rather than focusing solely on individual steps. Specifically:
>      - **Pathfinding-Based Retrieval Method:** This method finds the reasoning path in the KG that minimizes the overall semantic distance to the LLM-generated candidate paths. It ensures that even if the first relation in the reasoning path is not the most semantically similar, our algorithm can still identify the correct reasoning path if the complete path is close to the LLM’s plan. For example, in **Figure 2**, even if "CEO of" is not the most semantically similar relation in the first step, KARPA ensures that the entire path "CEO of → wife of CEO", is identified as the promising path to the answer entity.
>      - **Heuristic Value-Based Retrieval Method:** This method addresses the limitation of the pathfinding-based retrieval method, where the semantic cost between paths with different lengths cannot be calculated (please refer to **Weakness 2**). By incorporating a heuristic cost function that considers the semantic similarities between **entire paths**, KARPA ensures better alignment between the LLM-generated paths and the actual reasoning paths within the KG.
>      - **Steady Improvement:** In our paper, results in **Figure 3** and **Table 4** show a steady improvement in performance across the three retrieval methods: KARPA-B (beam search-based), KARPA-P (pathfinding-based), and KARPA-H (heuristic value-based). These results validate the effectiveness of KARPA in overcoming suboptimal answers caused by local optima.

---

> ### Author Response · Authors · 2024-11-26
> **Response to Reviewer Uom3 (2)**
>
> > **W2. Extend Relation Paths with Different Lengths**
>
> Your understanding is correct, the heuristic value-based retrieval method indeed addresses
> scenarios where the length of candidate paths generated by the LLM differs from the true
> reasoning paths in the KG. This situation arises often due to the variability in how LLMs
> reason and the structure of KGs.
>
> For example, as illustrated in **Figure 2** of our paper, the correct answer entity requires traversing the
> reasoning path "CEO of → wife of CEO" in the KG. However, during the re-planning step, the LLM
> utilize top-K relevant relations to form complete reasoning chains, which might fail to include
> the relation "CEO of" in its candidate paths, possibly generating a
> **single-relation candidate path** "wife of CEO".
>
> In such cases, standard **pathfinding-based retrieval method** that rely on
> direct semantic similarity between paths of equal lengths would fail to identify the
> correct reasoning path, as they cannot compute the **similarity between paths of differing
> lengths**. However, the **heuristic value-based retrieval method** can handle this mismatch by
> **treating the entire path as a whole** and calculating the cost between paths with different
> lengths by considering the semantic similarities between complete paths.
>
> To validate this, we restrict the retrieval step to use only **single-relation candidate paths**
> provided by the LLM during re-planning step, and compare the performance of the heuristic
> value-based retrieval method (KARPA-H) with the pathfinding-based retrieval method (KARPA-P)
> using GPT-4o-mini.
>
> | Candidate Path         | Method  | Accuracy (WebQSP) | Hit@1 (WebQSP) | F1 (WebQSP) | Accuracy (CWQ) | Hit@1 (CWQ) | F1 (CWQ) |
> |------------------------|---------|-------------------|----------------|-------------|----------------|-------------|----------|
> | Original Paths         | KARPA-P | 66.0              | 81.2           | 63.8        | 61.0           | 64.5        | 53.4     |
> | Original Paths         | KARPA-H | 72.3              | 86.4           | 67.2        | 64.6           | 67.7        | 55.1     |
> | Single-Relation Paths  | KARPA-P | 63.6              | 77.3           | 60.7        | 40.5           | 43.9        | 39.3     |
> | Single-Relation Paths  | KARPA-H | 71.4              | 85.5           | 68.9        | 55.1           | 59.6        | 47.4     |
>
> The results in the Table demonstrate that the heuristic value-based retrieval method
> **outperforms** pathfinding-based retrieval methods in such scenarios, as it effectively addresses the
> semantic similarity issues that arise from differing path lengths. Moreover, as the questions
> in the CWQ dataset generally **require longer reasoning paths** compared to WebQSP (detailed dataset
> parameters can be found in the **Datasets section** of the appendix), both methods exhibit a more
> significant decline in various metrics on CWQ. However, the heuristic value-based retrieval
> method shows a **less pronounced drop** compared to pathfinding-based retrieval methods, further
> demonstrating its superiority. We will add the experiment into our paper to further support
> this claim.

---

> ### Author Response · Authors · 2024-11-26
> **Response to Reviewer Uom3 (3)**
>
> > **W3. Performance on KGs Outside the Training Scope**
>
> Thank you for raising this question. The LLMs used in our experiments might have seen parts
> of WebQSP and CWQ datasets during pre-training. However, KARPA is designed to **minimize
> reliance on the LLM's internal knowledge** and instead **leverage its reasoning and planning
> capabilities** to construct logical reasoning chains.
>
> In KARPA, to correctly answer a question, the LLM must construct a reasoning path that
> aligns with the KG’s structure and leads to the correct answer entity. This means KARPA
> **emphasizes reasoning capability over stored knowledge**, making it capable of handling KGs
> that the LLM has not seen before.
>
> To further validate this, we compare KARPA with Chain-of-Thought (CoT) reasoning, where
> the LLM **directly relies on its internal knowledge** to answer questions. Using smaller-scale LLMs
> such as Qwen2.5-7B and Qwen2.5-14B (with limited stored knowledge), we observe that CoT
> performance drops significantly on KGQA tasks while KARPA maintains strong performance.
>
> | Base-model  | Method   | Accuracy (WebQSP) | Hit@1 (WebQSP) | F1 (WebQSP) | Accuracy (CWQ) | Hit@1 (CWQ) | F1 (CWQ) |
> |-------------|----------|-------------------|----------------|-------------|----------------|-------------|----------|
> | Qwen2.5-7B  | CoT      | -                 | 41.5           | -           | -              | 28.3        | -        |
> |             | KARPA    | 65.6              | 79.2           | 58.6        | 47.6           | 52.7        | 38.8     |
> |             | **Gain** | -                 | +37.7          | -           | -              | +24.4       | -        |
> | Qwen2.5-14B | CoT      | -                 | 49.6           | -           | -              | 31.2        | -        |
> |             | KARPA    | 72.6              | 84.1           | 65.0        | 51.5           | 57.9        | 41.6     |
> |             | **Gain** | -                 | +34.5          | -           | -              | +26.7       | -        |
> | Qwen2.5-72B | CoT      | -                 | 56.9           | -           | -              | 40.5        | -        |
> |             | KARPA    | 73.2              | 86.0           | 64.5        | 61.1           | 63.6        | 52.7     |
> |             | **Gain** | -                 | +29.1          | -           | -              | +23.1       | -        |
>
> The results highlight KARPA's ability to operate effectively on unseen KGs by focusing on
> reasoning and planning rather than leveraging the LLM’s pre-existing knowledge. We will add
> these results into our revised paper.
>
> We hope that our responses have addressed your concerns and clarified our approach.
> Should you have any further feedback or suggestions, we would be glad to address them. If our responses have resolved your concerns, we humbly ask your consideration in revisiting the rating. Thank you once again for your valuable and insightful review!

---

> > ### Comment · Reviewer_Uom3 · 2024-11-28
> > **Feedback from Reviewer Uom3**
> >
> > Thanks for the authors' feedback. In my opinion, combining KG with LLMs is to exploit different knowledge in them. However, the knowledge in current KGs and dataset seems to have been already learned in current LLMs. As a result, there is a data leak problem. It is very hard to say that the knowledge from KGs is really useful for the reasoning process. So experiments on unseen KGs and generalization are needed and very important.

---

### Official Review · Reviewer_kVb3 · 2024-11-03

**Soundness:** 2
**Presentation:** 2
**Contribution:** 2
**Rating:** 5
**Confidence:** 4

**Summary:**

This paper proposes KARPA, a training-free KGQA method that utilizes LLMs to reason over KGs. KARPA uses a pre-planning step to extract candidate relation paths in a three-stage manner (initial planning, relation extraction, and re-planning) and ranks top candidates in the later reasoning step. The main difference between KARPA and previous LLM x KGQA methods (e.g. ToG and Pangu) is that KARPA is more efficient in LLM usage as it avoids the stepwise interaction between LLM and KG, and yet achieves superior performance. Regarding the experimental results, KARPA consistently outperforms baseline methods on WebQSP and CWQ datasets.

Despite these strengths, KARPA also exhibits several limitations. First, KARPA relies heavily on LLM's planning and reasoning abilities since it needs LLM to propose candidate relation paths or rank candidates in one single call, which may be harder to succeed when the candidate subgraph is complex and large. Second, in the pre-planning stage, KARPA uses a traditional sentence transformer to embed relations and calculate similarity, which may fail in multilingual scenarios (for example, the initial relations proposed by LLM is English, but the relation surface form in KG is Chinese). Overall, I may doubt the performance of KARPA on more complex KG (larger subgraphs, more relations) and multilingual scenarios.

**Strengths:**

- The proposed method is well-motivated and reasonable under certain scenarios.
- The presented experimental results are strong.

**Weaknesses:**

- KARPA relies heavily on LLM's planning and reasoning abilities since it needs LLM to propose candidate relation paths or rank candidates in one single call, which may be harder to succeed when the candidate subgraph is complex and large.
- In the pre-planning stage, KARPA uses a traditional sentence transformer to embed relations and calculate similarity, which may fail in multilingual scenarios (for example, the initial relations proposed by LLM is English, but the relation surface form in KG is Chinese).
- Some experiment details are not clear enough. See questions.

**Questions:**

- In lines 408-409, what does "For each LLM, we sample 300 KGs instances across the datasets to optimize computational cost" mean?
- It seems that KARPA only requires 3 LLM calls: initial planning, re-planning, and reasoning. Can you explain more about the results in Table 3?
- Instead of call frequency, token usage is a more effective measure for illustration. Do you have a token usage comparison between KARPA and ToG?

---

> ### Author Response · Authors · 2024-11-26
> **Response to Reviewer kVb3 (1)**
>
> We sincerely thank you for your constructive feedback.
> Below, we provide detailed responses to your questions and concerns.
>
> ### Weakness:
>
> > **W1. Reliance on LLM Planning and Reasoning Abilities**
>
> Thank you for raising this concern. While our approach indeed leverages the LLM’s planning
> and reasoning abilities, KARPA significantly **reduces** this reliance compared to other methods
> that perform direct inference over knowledge graphs (KGs) step-by-step with LLMs.
> Traditional inference-based methods like ToG require the LLM to repeatedly interact with the
> KG, selecting relations from hundreds or even thousands of **adjacent relations** at each step.
> In contrast, KARPA uses an embedding model to extract the top-K (K=30 in our paper)
> **most relevant relations** from the KG. These relations are then presented to the LLM, enabling it
> to **construct logically coherent relation paths** that directly target the answer entities.
>
> Moreover, we believe that allowing the LLM to generate complete reasoning chains aligns better
> with **human-like reasoning processes**. In contrast, other inference-based methods fragment the
> reasoning process into multiple steps, underutilizing the LLM’s global planning capabilities.
>
> Our experiments further validate this. Here, we provide additional results using **Qwen2.5-7B**
> and **Qwen2.5-14B**, which are weaker in planning and reasoning capabilities:
>
> | Base-model  | Method | Accuracy (WebQSP) | Hit@1 (WebQSP) | F1 (WebQSP) | Accuracy (CWQ) | Hit@1 (CWQ) | F1 (CWQ) |
> |-------------|--------|-------------------|----------------|-------------|----------------|-------------|----------|
> | Qwen2.5-7B  | CoT    | -                 | 41.5           | -           | -              | 28.3        | -        |
> |             | ToG    | 24.6              | 30.2           | 21.9        | 22.4           | 25.8        | 20.2     |
> |             | KARPA  | 65.6              | 79.2           | 58.6        | 47.6           | 52.7        | 38.8     |
> | Qwen2.5-14B | CoT    | -                 | 49.6           | -           | -              | 31.2        | -        |
> |             | ToG    | 45.0              | 55.9           | 42.7        | 31.2           | 36.6        | 29.5     |
> |             | KARPA  | 72.6              | 84.1           | 65.0        | 51.5           | 57.9        | 41.6     |
>
> Even with these smaller LLMs, KARPA outperforms CoT and ToG, demonstrating more stable
> performance across varying reasoning capabilities. This highlights KARPA’s robustness
> and its **reduced dependence** on the LLM’s planning abilities.
>
> > **W2. Handling Multilingual Scenarios**
>
> Thank you for pointing out this important question. We agree that multilingual scenarios
> require special consideration. However, this challenge can be effectively addressed by
> using **multilingual embedding models**. For instance, in a multilingual setting, we test KARPA
> with **paraphrase-multilingual-MiniLM-L12-v2**, a multilingual embedding model. In this experiment,
> we use GPT-4o-mini to generate relation paths in Chinese, and then use the multilingual embedding
> model to calculate the semantic similarity between the candidate paths and paths in the KG.
>
> | Language        | Accuracy (WebQSP) | Hit@1 (WebQSP) | F1 (WebQSP) | Accuracy (CWQ) | Hit@1 (CWQ) | F1 (CWQ) |
> |-----------------|-------------------|----------------|-------------|----------------|-------------|----------|
> | English-English | 74.1              | 85.3           | 68.3        | 65.3           | 69.5        | 55.4     |
> | Chinese-English | 74.6              | 84.5           | 67.6        | 63.1           | 68.0        | 54.2     |
>
> These results demonstrate that with a multilingual embedding model, KARPA performs effectively
> across languages, maintaining its robustness. They also indicate that language variations
> do not significantly impact KARPA’s performance.
>
> ### Questions:
>
> > **Q1. Clarity of Experimental Details**
>
> We appreciate your comment. Specifically, this sentence means that we randomly select 300
> knowledge graphs (KGs) from each of the KGQA datasets (WebQSP, CWQ) to evaluate the performance
> of KARPA with different LLMs, aiming to save on computational costs. We will clarify this
> sentence in our revised paper.

---

> ### Author Response · Authors · 2024-11-26
> **Response to Reviewer kVb3 (2)**
>
> > **Q2. Explanation of Table 3 Results**
>
> Your understanding is correct, KARPA only utilizes LLMs in three steps: initial planning,
> re-planning, and reasoning. However, the re-planning step often generates **multiple candidate
> paths**, especially for complex questions or when there are multiple topic entities.
> Each of these candidate paths is matched to paths within the KG using semantic similarity
> to retrieve the most relevant reasoning paths. In the reasoning step, the top-K retrieved
> paths of each candidate paths are provided to the LLM in batches to generate the final answers.
> As the complexity of the query increases (e.g., in the CWQ dataset), the number of topic
> entities and candidate paths also increases. Consequently, the number of LLM calls
> during the reasoning step rises.
>
> In **Table 3** of our paper, we observe that the CWQ dataset requires more LLM calls compared to WebQSP due to
> its more complex query logic. However, compared to methods that relies on direct interation
> between LLMs and KGs such as ToG, where LLM call frequency increases significantly with question
> complexity, KARPA demonstrates much more stable scaling. For instance, in Table 3,
> ToG requires an average of **3.1
> additional calls** for the CWQ dataset, while KARPA requires only **0.5 additional calls** when using
> GPT-4o and GPT-4. We will include additional details in the Experiments section to
> better explain the LLM call frequency in Table 3.
>
> > **Q3. Token Usage Comparison**
>
> Thank you for suggesting this additional evaluation metric. We agree that token usage is an
> important measure of efficiency. Here, we provide a token usage comparison between KARPA and ToG
> using the tokenizer of GPT-4o-mini.
>
> | Method | Input tokens/KG (WebQSP) | Output tokens/KG (WebQSP) | Input tokens/KG (CWQ) | Output tokens/KG (CWQ) |
> |--------|--------------------------|---------------------------|-----------------------|------------------------|
> | ToG    | 6351.5                   | 1836.5                    | 7935.7                | 2931.6                 |
> | KARPA  | 2465.9                   | 1492.3                    | 3612.1                | 2267.1                 |
>
> The results show that KARPA **significantly reduces** both input and output token usage compared
> to ToG, which means we have not only lowered the **reasoning complexity** for the LLM but also
> saved on the **computational costs** of the LLM, further demonstrating the superiority of KARPA.
>
> We would like to learn if our response addresses your concerns and questions, and we invite any additional feedback or thoughts for improving our paper. If you feel that our responses resolve the issues raised, we would be grateful if you could consider reflecting this in the evaluation. We would be happy to address any further concerns or questions.

---

### Official Review · Reviewer_Es96 · 2024-11-03

**Soundness:** 2
**Presentation:** 2
**Contribution:** 2
**Rating:** 5
**Confidence:** 4

**Summary:**

The paper proposes Knowledge graph Assisted Reasoning Path Aggregation (KARPA), a novel framework that harnesses the global planning abilities of LLMs for efficient and accurate KG reasoning on KGs. KARPA operates through a three-step process: pre-planning, retrieving, and reasoning.

**Strengths:**

The idea is reasonable. The paper clearly describes their model.

**Weaknesses:**

1. The comparison in Table1 is unfair, the baseline all use 3-7B LLM but the proposed model uses GPT. The proposed model heavily depends on the output of the LLMs. Whether the authors have tried the effect of using 7B LLMs.

2. The claimed "training-free" contribution comes from the pre-trained LLMs and relies on the ability of LLMs. Therefore, KARPA lacks substantive technical contributions.

3. For a specific problem, what if the path given by LLM does not match the knowledge graph? In the face of diverse query problems, how can the output of LLM remain effective? In this sense, ToG may be more realistic and efficient.

**Questions:**

1. What is the difference between KARPA and other prompt-based methods? It seems they are just involved in the change of input.

2. How to control the quality of LLM output, since it is the most important in this kind of methods. What if LLM outputs some meaningless paths or hallucinations?

3. How should KARPA work when KG is very dense (eg, too many paths)? It is hard to prove that the selection of Embedding models is better than the selection of LLM.

---

> ### Author Response · Authors · 2024-11-26
> **Response to Reviewer Es96 (1)**
>
> We would like to thank you for the constructive feedback on our paper. Below, we address
> each of your concerns in detail.
>
> ### Weakness:
>
> > **W1. KARPA with Smaller LLMs**
>
> Thank you for highlighting this concern. In Table 1, we classify baselines into two groups:
> **training-based methods** and **direct inference over knowledge graphs (KGs)** with LLMs.
> Smaller LLMs (3–7B) are typically employed in training-based KGQA methods,
> while more powerful LLMs are generally required for inference-based approaches like ToG and KARPA,
> which rely on reasoning capabilities of LLMs.
>
> To address your concern, we conduct additional experiments with **Qwen2.5-7B** and **Qwen2.5-14B**
> as the LLM backbones for KARPA. The results demonstrate that KARPA consistently outperforms
> stepwise direct inference baselines such as ToG, even when using smaller LLMs.
> This reinforces the robustness and adaptability of our method across different LLM scales.
>
> | Base-model  | Method | Accuracy (WebQSP) | Hit@1 (WebQSP) | F1 (WebQSP) | Accuracy (CWQ) | Hit@1 (CWQ) | F1 (CWQ) |
> |-|-|-|-|-|-|-|-|
> | Qwen2.5-7B  | CoT    | -                 | 41.5           | -           | -              | 28.3        | -        |
> |             | ToG    | 24.6              | 30.2           | 21.9        | 22.4           | 25.8        | 20.2     |
> |             | KARPA  | 65.6              | 79.2           | 58.6        | 47.6           | 52.7        | 38.8     |
> | Qwen2.5-14B | CoT    | -                 | 49.6           | -           | -              | 31.2        | -        |
> |             | ToG    | 45.0              | 55.9           | 42.7        | 31.2           | 36.6        | 29.5     |
> |             | KARPA  | 72.6              | 84.1           | 65.0        | 51.5           | 57.9        | 41.6     |
>
> > **W2. Technical Contributions**
>
> We would like to clarify the technical contributions of KARPA. While many training-free
> approaches depend on pre-trained LLMs, our innovation lies in the unique utilization of
> LLM reasoning and planning capabilities. Specifically:
>
> - As mentioned in our paper, traditional inference-based approaches adopts a step-by-step
> relation selection approach using LLMs, which often fails to construct globally coherent
> reasoning chains. In contrast, KARPA enables LLMs to leverage all semantically relevant
> relations in the KG to construct comprehensive and logically sound reasoning paths.
>
> - We introduce three novel reasoning path retrieval mechanisms that combine semantic similarity
> with pathfinding algorithms, addressing the local optimality problem in prior methods.
>
> - As highlighted in our paper, KARPA achieves state-of-the-art (SOTA) performance across
> different KGQA benchmarks, with significantly fewer interactions between the LLM and KG.
>
> By identifying and addressing critical limitations in existing methods, we believe our
> work makes a meaningful contribution to the field, even without training a new model.
>
> > **W3. Handling Mismatches Between LLMs and KGs**
>
> We appreciate the reviewer’s insightful observation, which touches on a critical issue that we
> have carefully considered in the design of KARPA.
> As outlined in our paper, we have decomposed the pre-planning step into three substeps:
> initial planning, relation extraction, and re-planning (see **Figure 2** in the paper, where
> Pre-Planning is detailed on the left). In **initial planning step**, the LLM generates relation
> paths without knowledge of the KG. These paths may indeed lack alignment with the KG.
> To address this, we extract semantically relevant relations from the KG in relation
> extraction step. In **re-planning step**, the extracted relations are then used to refine
> and re-plan reasoning paths. This guarantees that the candidate paths are both **logically
> coherent** and **aligned with the KG**.
>
> To quantify the impact of the re-planning step, we provide an ablation study below.
> Results show that **removing the re-planning step** significantly reduces KARPA's performance,
> confirming its importance. We use GPT-4o-mini as our base-model.
>
> | Pre-planning Step| Accuracy (WebQSP) | Hit@1 (WebQSP) | F1 (WebQSP) | Accuracy (CWQ) | Hit@1 (CWQ) | F1 (CWQ) |
> | -|-|-|-|-|-|-|
> | Origin | 72.3| 86.4| 67.2 | 64.6| 67.7 | 55.1 |
> | Remove Re-Planning Step | 64.1| 79.6 | 61.5 | 54.3| 59.5 | 47.1     |
>
> Additionally, in the retrieval step, KARPA employs semantic similarity as the cost
> function for pathfinding algorithms. This ensures that the final reasoning paths selected
> not only **exist in the KG** but are also **semantically closest** to the paths generated by
> the LLM, thereby maintaining the validity of the LLM's output across
> diverse query problems.
>
> **Regarding efficiency**: as shown in **Tables 2 and 3** in our paper, when tested with the same LLM on the
> same datasets, KARPA significantly outperforms ToG both in terms of accuracy and interaction
> efficiency. These results underscore KARPA’s practicality and computational efficiency.

---

> ### Author Response · Authors · 2024-11-26
> **Response to Reviewer Es96 (2)**
>
> ### Questions:
>
> > **Q1. Differences Between KARPA and Other Prompt-Based Methods**
>
> We would like to highlight the distinctions of KARPA compared to existing prompt-based methods.
> Most LLM-based training-free KGQA methods only alter the input prompts, but the information
> provided to LLMs by KARPA's prompts is unique. As noted in our paper, existing prompt-based
> methods typically allow the LLM to select next relations from the current entity's
> neighboring relations, potentially leading to incomplete or inefficient reasoning paths.
> KARPA enables the LLM to **directly construct complete and logically coherent reasoning paths**,
> without the need for multiple interactions between the LLM and the KG.
> Our method not only avoids the issue of the LLM getting trapped in locally optimal solutions
> during direct exploration of the KG but also reduces the number of LLM-to-KG interactions.
>
> Additionally, unlike other prompt-based methods that rely on beam search to handle dense
> KGs, we propose several novel retrieval strategies based on semantic similarity and pathfinding
> algorithms. These methods address the non-optimal solutions arising from beam search and further
> reduce the LLM-to-KG interactions. Through these unique technical innovations, KARPA achieves
> SOTA performance in KGQA tasks, as demonstrated in our experiments.
>
> > **Q2. Controlling the Quality of LLM Output**
>
> You raise an important point, as the quality of LLM outputs is indeed critical in our methods.
> In our paper, KARPA employs several mechanisms to address this issue. We use an semantic embedding
> model during both the pre-planning and retrieval stages to ensure that LLM outputs align with
> the KG structure. The embedding model effectively acts as a **corrective mechanism** to prevent
> hallucinations:
> - In the re-planning step of pre-planning, the LLM refines its initial paths
> using relations extracted from the KG based on their semantic similarity to the LLM’s output.
> This ensures that the final candidate paths are both logically coherent and grounded in the KG.
> - In the retrieval stage, we employ semantic similarity combined with pathfinding algorithms
> to select paths that are both aligned with the KG and most semantically similar to the
> LLM-generated paths.
>
> These mechanisms eliminate the risk of meaningless paths or hallucinations,
> ensuring that KARPA outputs robust and trustworthy reasoning paths.
>
> > **Q3. Performance with Dense Knowledge Graphs**
>
> We appreciate your focus on of KARPA for dense KGs. When the KG is dense, methods that rely
> on step-by-step interactions between the LLM and KG (selecting with LLMs) face significant
> challenges. Specifically, the LLM must select the next relations from **hundreds or even thousands**
> of adjacent relations at each step, and repeat this process until the answer entities are found.
> This results in a high computational burden, and also fails to leverage the LLM’s global planning
> capabilities. In contrast, KARPA’s pre-planning phase allows the LLM to construct **complete
> reasoning chains** using top-K relevant relations (K = 30 in our paper) from the KG. This significantly
> reduces the LLM’s burden in dense KGs and fully exploits the LLM's global planning ability.
> Here, we provide a comparison of the average number of input tokens between ToG and KARPA using the
> tokenizer of GPT-4o-mini:
>
> | Method | Input tokens (WebQSP) | Input tokens (CWQ) |
> |--------|-----------------------|--------------------|
> | ToG    | 6351.5                | 7935.7             |
> | KARPA  | 2465.9                | 3612.1             |
>
> Additionally, in the retrieval phase, we apply three **semantic similarity-based retrieval methods**,
> which further reduce the number of LLM-KG interactions while avoiding hallucinations. In our paper,
> we compare our method with ToG (selecting with LLMs) in **Tables 2 and 3**, and KARPA demonstrates
> superior performance in both accuracy and efficiency. These results also demonstrate that embedding
> models can indeed outperform LLMs in dense KG scenarios with better accuracy and efficiency.
>
>
> Thank you again for your thoughtful comments and feedbacks. If our responses have satisfactorily
> addressed your concerns, we humbly ask the reconsideration of the rating. We remain open and eager to address any further concerns or questions you may have, please feel free to share any additional feedback.

---

> > ### Comment · Reviewer_Es96 · 2024-11-29
> >
> > Thank you for your response!
> >
> > I have read your reply and will maintain my initial scores.

---

> ### Author Response · Authors · 2024-11-29
> **Response to Reviewer Es96 (3)**
>
> Thank you for your response and for taking the time to review our work thoroughly!
>
> We sincerely believe that our response has comprehensively addressed the questions and concerns you raised. We kindly ask you to consider increasing the score. If there are any additional questions or issues that remain unresolved, we would be more than happy to provide further clarifications.
>
> Once again, we deeply appreciate your time and thoughtful feedback.

---

### Official Review · Reviewer_fYqD · 2024-11-09

**Soundness:** 2
**Presentation:** 3
**Contribution:** 2
**Rating:** 3
**Confidence:** 4

**Summary:**

The framework operates through a three-step process: pre-planning, retrieving, and reasoning. KARPA is designed to address the limitations of existing LLM-based KGQA methods by fully utilizing the global planning and reasoning capabilities of LLMs without requiring stepwise traversal or additional training. The paper claims that KARPA achieves state-of-the-art performance in KGQA tasks, offering both high efficiency and accuracy.

**Strengths:**

- The three-step process of pre-planning, retrieving, and reasoning is a easy-to-fellow approach to leverage external knowledge sources for enhancing LLMs.

- KARPA's training-free nature is an advantage, as it allows for easy integration with various LLM architectures without the need for fine-tuning or pre-training on specific KGs, which can be time-consuming and resource-intensive.

**Weaknesses:**

- To my knowledge, The proposed method lacks novelty; employing an agent-based approach for KGQA tasks is not particularly innovative.  For instance, RoG recently introduced a planning-retrieval-reasoning framework for KGQA, and  this manuscript is the expansion of the planning phase in RoG's framework into two processes: pre-planning and re-planning.

- Secondly, the authors claim in their motivation that "Pre-training or fine-tuning the LLM for KGQA, which is prone to hallucinations and struggles to adapt to unseen KGs without an extensive training process." I believe this is a point worth discussing. Taking RoG as an example again, it did employ instruction-tuning for the LLM, but I think such instruction-tuning for LLMs possesses a certain ability to learn from few samples, meaning that generalization can be reflected in the mitigation of hallucinations. Of course, I also feel that this is worth experimenting with and discussing further.

- Considering the forward-looking nature of RoG in initially proposing the planning-retrieval-reasoning approach, it is somewhat unfair that this manuscript did not use the same large model as a backbone in its comparison with RoG (Table 1).

- There is a noticeable absence of robust agent-like methods, such as Interactive-KBQA, which directly perform inference over KGs with LLMs.

**Questions:**

- Could the authors elaborate on the choice of semantic embedding models and how different models affect the performance of KARPA?


- How does KARPA perform on the strict EM (Etract Match) metric, as many baselines have used this metric for experiments.

---

> ### Author Response · Authors · 2024-11-26
> **Response to Reviewer fYqD (1)**
>
> Thank you for your thorough review and valuable insights. Below, we address your concerns
> and provide additional clarifications and experimental results to strengthen our work.
>
> ### Weakness:
>
> > **W1. Novelty of KARPA**
>
> We acknowledge reasoning on graphs (RoG) as a foundational contribution to the domain of LLM-based KGQA. However,
> KARPA differs significantly from RoG in several key aspects:
> - Unlike RoG, which requires
> a training phase to inject KG knowledge into the LLM, KARPA is **training-free**. This avoids the
> extensive computational cost and potential biases introduced during training, making KARPA
> both efficient and adaptable to unseen KGs.
> - In KARPA, the initial-planning and
> re-planning steps ensure that the LLM uses all relevant relations from the KG to construct
> reasoning paths. This minimizes hallucinations and generates reasoning paths that are both
> complete and logically consistent.
> - We propose three retrieval methods based on
> semantic similarity and pathfinding algorithms, including the heuristic value-based retrieval
> method, which addresses scenarios with paths of differing lengths. These methods are
> specifically designed to mitigate hallucinations, which is a common challenge in RoG’s
> framework (see **Weakness 2**).
> - Even when using the same backbone LLM as RoG, our
> approach achieves better results due to its robust retrieval strategies (see **Weakness 3**).
>
> KARPA introduces significant improvements in methodology, experimental results, and practical
> usability. Its plug-and-play nature allows seamless integration into existing systems,
> further enhancing its value. We are confident that our work makes a meaningful contribution to
> the field of LLM-based KGQA.
>
> > **W2. Hallucinations and Adaptation to Unseen KGs**
>
> We agree that RoG’s instruction-tuning improves LLMs' ability to learn from a few samples,
> providing some generalization capabilities.
> However, when the KG or corresponding questions
> change, instruction-tuned LLMs may generate reasoning paths that only partially match the KG,
> leading to failure in extracting the correct reasoning paths. In this case, frequent
> re-tuning is required to adapt to new KGs or modified questions, which can be computationally
> expensive and impractical in real-world applications.
>
> Our **semantic similarity-based retrieval methods** address this challenge by extracting the most semantically similar and accurate paths within the KG. To validate this,
> we conduct an experiment using GPT-4o-mini on a **modified version** of the WebQSP dataset.
> Specifically, we slightly alter the questions in WebQSP while preserving their original meaning
> (using the prompt: “Please revise the question to make it more clear, but the original
> meaning of the question and the corresponding answers remain unchanged.”). For example, consider the question “What type of cancer did Eva Peron have?” revised to “What type of cancer was Eva Peron diagnosed with?”. In this case, the instruction-tuned LLaMa2-Chat-7B generates incorrect reasoning paths, missing the correct answer "Cervical cancer".
>
> We test RoG with instruction-tuned LLaMa2-Chat-7B in planning step and GPT-4o-mini for reasoning. In KARPA, we use
> GPT-4o-mini for both pre-planning and reasoning steps.
>
> |Question|Method|Accuracy|Hit@1|F1|Method|Accuracy|Hit@1|F1|
> |-|-|-|-|-|-|-|-|-|
> |Origin|RoG|67.6|84.1|69.7|KARPA|73.1|85.4|68.1|
> |Revised|RoG|63.5|74.3|64.1|KARPA|72.6|84.5|68.9|
> |Variation|RoG|-4.1|-9.8|-5.6|KARPA|-0.5|-0.9|+0.8|
>
> The results show that KARPA’s performance remains consistent and robust to
> question modifications, while RoG’s performance drops due to path mismatches. This further highlights the advantage of KARPA's **training-free** framework, maintaining superior robustness and adaptability across all KGs.
>
> > **W3. Comparison with RoG Using the Same Backbone LLM**
>
> We appreciate RoG's contribution to this domain and its forward-looking framework, and we will
> emphasize RoG’s contributions to the field of KGQA in our related work section. Here, we
> conduct an additional experiment using **instruction-tuned** LLaMa2-Chat-7B as the backbone LLM for
> both KARPA and RoG, while using **untrained** Qwen2.5-7B and Qwen2.5-14B for final answer reasoning in
> both methods.
>
> |Base-model|Method|Accuracy (WebQSP)|Hit@1 (WebQSP)|F1 (WebQSP)|Accuracy (CWQ)|Hit@1 (CWQ)|F1 (CWQ)|
> |-|-|-|-|-|-|-|-|
> |LLaMa2-7B + Qwen2.5-7B|RoG|54.5|73.8|57.2|38.6|43.5|35.8|
> ||KARPA|66.4|82.7|63.6|54.1|59.2|46.3|
> |LLaMa2-7B + Qwen2.5-14B|RoG|58.7|77.2|60.9|43.9|48.0|42.5|
> ||KARPA|69.8|84.2|67.4|55.0|60.4|47.2|
>
> Even with the same backbone LLM, KARPA’s retrieval methods successfully extract more accurate
> reasoning paths, leading to higher accuracy in final answers. We will add the results into our revised paper.

---

> ### Author Response · Authors · 2024-11-26
> **Response to Reviewer fYqD (2)**
>
> > **W4. Inclusion of Interactive-KBQA**
>
> Thank you for highlighting this paper. We agree that Interactive-KBQA is a significant agent-like method
> that should be included into our discussion of related work. Interactive-KBQA shares similarities with
> ToG as both **approaches rely on direct, step-by-step interaction** between LLMs and KGs to infer answers.
> However, these methods face challenges such as: (1) Sequential LLM-KG interactions result in high computational costs,
> especially for complex queries or large KGs. (2) The step-by-step reasoning process often
> leads to suboptimal paths due to the lack of holistic context.
>
> In contrast, KARPA eliminates the need for iterative interaction by directly generating a complete
> reasoning path based on relations extracted from the KG. Our approach significantly reduces the computational
> cost for LLMs and improves the logical coherence of reasoning paths. To further substantiate KARPA’s advantages,
> we conduct an additional experiment comparing KARPA with Interactive-KBQA, using GPT-4-turbo as the backbone
> LLM. The results of Interactive-KBQA are cited from its paper.
>
> | Method           | 1-hop F1 (WebQSP) | 2-hop F1 (WebQSP) | Overall F1 (WebQSP) | RHits@1 (WebQSP) | Overall F1 (CWQ) |
> |------------------|-------------------|-------------------|---------------------|------------------|------------------|
> | Interactive-KBQA | 69.99             | 72.41             | 71.20               | 72.47            | 49.07            |
> | KARPA            | 74.21             | 72.97             | 73.78               | 74.14            | 61.45            |
>
> The results show that KARPA outperforms Interactive-KBQA on WebQSP and CWQ datasets with GPT-4-turbo. We
> will include Interactive-KBQA in our revised paper.
>
> ### Questions:
>
> > **Q1. Impact of Different Embedding Models**
>
> Thank you for this suggestion. We conduct additional experiments comparing different
> embedding models to evaluate their impact on KARPA’s performance with GPT-4o-mini.
>
> | Embedding Model                                | Accuracy (WebQSP) | Hit@1 (WebQSP) | F1 (WebQSP) | Accuracy (CWQ) | Hit@1 (CWQ) | F1 (CWQ) |
> |------------------------------------------------|-------------------|----------------|-------------|----------------|-------------|----------|
> | all-MiniLM-L6-v2 (~86MB)                       | 72.3              | 86.4           | 67.2        | 64.6           | 67.7        | 55.1     |
> | all-mpnet-base-v2 (~417MB)                     | 74.5              | 86.1           | 68.6        | 64.1           | 68.3        | 53.7     |
> | paraphrase-multilingual-MiniLM-L12-v2 (~448MB) | 74.1              | 85.3           | 68.3        | 65.3           | 69.5        | 55.4     |
>
> In the table, all-MiniLM-L6-v2 is the embedding model used in our paper, with a size of
> approximately 86MB. all-mpnet-base-v2, a more powerful embedding model, is around 417MB in size.
> Lastly, paraphrase-multilingual-MiniLM-L12-v2, which supports multiple languages, has a size of
> approximately 448MB.
> The results show that different embedding models have a **minimal** impact on the results of KARPA.
> We will add these experimental results to the revised manuscript to provide a more comprehensive
> discussion on embedding model choices.
>
> > **Q2. Performance on the Exact Match (EM) Metric**
>
> Thank you for reminding us about the exact match (EM) metric. KARPA follows the setting of RoG and ToG,
> and we did not include EM scores in our original submission. We now provide additional results
> on KARPA’s EM metric.
>
> | Base-model | Method  | EM (WebQSP) | EM (CWQ) |
> |------------|---------|-------------|----------|
> | GPT-4o     | ToG     | 39.5        | 37.6     |
> |            | KARPA   | 44.6        | 41.3     |
> | GPT-4      | ToG     | 43.1        | 40.9     |
> |            | KARPA   | 51.7        | 47.2     |
>
> The results demonstrate that KARPA achieves higher EM scores compared to ToG,
> showing its effectiveness in accurately extracting reasoning paths and final answers.
> We will include the results in our revised paper.
>
> We are grateful for your detailed feedback, which has allowed us to strengthen our work.
> If our rebuttal addresses your concerns and proves useful, we respectfully ask you to consider
> adjusting your review and scores.
> Thank you once again for your thoughtful review, and we look forward to your continued insights
> and feedback.

---

### Note · Authors · 2024-12-16

I have read and agree with the venue's withdrawal policy on behalf of myself and my co-authors.